# Magnitude and timescale of liquid water path adjustments to cloud droplet number concentration perturbations for nocturnal non-precipitating marine stratocumulus

Yao-Sheng Chen[1,2], Prasanth Prabhakaran[1,2], Fabian Hoffmann[3], Jan Kazil[1,2], Takanobu Yamaguchi[1,2], and Graham Feingold[2]

[1]Cooperative Institute for Research in Environmental Sciences, University of Colorado Boulder, Boulder, Colorado, USA
[2]NOAA Chemical Sciences Laboratory, Boulder, Colorado, USA
[3]Meteorologisches Institut, Ludwig-Maximilians-Universität München, Munich, Germany

**Correspondence:** Yao-Sheng Chen (yaosheng.chen@noaa.gov)

**Abstract.** Cloud liquid water path ($L$) adjusts to perturbations in cloud droplet number concentration ($N$) over time. We explore the magnitude and timescale of this adjustment in nocturnal non-precipitating marine stratocumuli using large eddy simulations of baseline conditions and aerosol seeding experiments for 22 meteorological conditions. The results confirm that the $L$ adjustment ($\delta L$) slope ($k$) is more negative for simulation pairs with relatively low $N$ and less negative for high $N$. Overall, $k$ is unlikely to be lower than $-0.4$ within 24 h since seeding starts, meaning the $L$ adjustment is unlikely to fully offset the brightening due to the Twomey effect. After seeding, the $\delta L$ becomes increasingly negative which can be characterized by an exponential convergence. This evolution is governed by a short timescale around 5 h and lasts for around 8–12 h. It is driven by the feedback between entrainment, $L$, and boundary layer (BL) turbulence. Other processes, including radiation, surface fluxes, and subsidence, respond to the seeding weakly. This short timescale is insensitive to the amount of seeding, making the evolution of $\delta L$ and some other deviations similar for different seeding amounts after appropriate scaling. The timescale of $k$ evolution is closely related to the $\delta L$ timescale and hence also short, while it could also be affected by the $\delta N$ evolution. The results are most relevant to conditions where seeding is applied to a large area of marine stratocumulus in well-mixed and overcast BL where shear is not a primary source of turbulence.

## 1 Introduction

Marine cloud brightening (MCB) has been proposed as a climate intervention strategy to mitigate global warming by taking advantage of the Twomey effect (Twomey, 1974, 1977): injecting aerosol particles into marine stratocumuli (henceforth "seeding") increases cloud droplet number concentration (denoted by $N$) and reduces cloud droplet size; with cloud water amount unchanged, this perturbation increases the cloud albedo to reflect more solar radiation back to space to cool the Earth (Latham, 1990; Feingold et al., 2024). The initial brightening occurs within a short space of time around 10–15 min as the seeded particles are transported vertically through the boundary layer (BL). Afterwards, the macroscopic properties of these clouds, including liquid water path (denoted with $L$) and cloud fraction, adjust to the changes in $N$ (Albrecht, 1989; Ackerman et al.,

2004; Bretherton et al., 2007), further modifying the albedo of a cloudy scene. The sign, the magnitude, and the timescale of these adjustments may affect the effectiveness of MCB.

In this study, we focus on the adjustment of $L$ to a perturbation in $N$ (abbreviated as "$L$ adjustment") that is initially caused by seeding. This adjustment can be characterized by a ratio

$$k = \frac{\delta l}{\delta n}, \tag{1}$$

where $l = \ln L$, $n = \ln N$, and $\delta$ indicates the difference between seeded and unseeded conditions. We refer to $k$ as the adjustment slope because it is the slope of the line segment connecting cloud states ($N$, $L$) with and without seeding in the $N$–$L$ plane on a log scale. It can be used to characterize the sensitivity of $l$ to $n$ ($\mathrm{d}l/\mathrm{d}n$) required to assess the susceptibility of cloud albedo to $N$ under idealized conditions (Platnick and Twomey, 1994; Bellouin et al., 2020). A positive $k$ indicates that the $L$ adjustment further brightens a cloudy scene in addition to the Twomey effect, while a negative $k$ indicates the opposite. In particular, a $k$ of $-0.4$ indicates that the $L$ adjustment exactly offsets the Twomey effect.

For non-precipitating marine stratocumuli, $k$ is likely negative but its magnitude is still uncertain. The negative sign comes from a few mechanisms. An initial increase in $N$ leads to smaller droplets. They evaporate faster during the mixing between the cloud and the free-troposphere (FT) air (Wang et al., 2003); they sediment slower, allowing more liquid near the cloud top to evaporate (Bretherton et al., 2007). The reduction of the drop size also suppresses the cloud-base precipitation, which may strengthen the turbulence (Lu and Seinfeld, 2005; Caldwell et al., 2005; Wood, 2007; Sandu et al., 2008). Smaller droplets may also enhance cloud-top longwave radiative cooling (Garrett et al., 2002; Petters et al., 2012; Igel, 2024). All these mechanisms suggest that an increase in $N$ enhances the cloud-top entrainment, which leads to more warming and drying of the BL and reduces $L$, although there is still debate on the dominant mechanism behind this enhancement (Igel, 2024). During the daytime, higher $N$ causes more absorption of solar radiation (Stephens, 1978; Boers and Mitchell, 1994; Petters et al., 2012), also contributing to a negative $k$, although the weaker absorption by lowered $L$ may limit this effect (Sandu et al., 2008; Zhang et al., 2024). The uncertainty in our understanding of the magnitude of $k$ can be seen in the wide range of values reported in previous studies, which used different methods and focused on different conditions (e.g., see a summary compiled in Figure 1 and Table S1 Glassmeier et al., 2021). The correlation between environmental conditions and $k$ is also under debate (e.g., Chun et al., 2023).

Several recent works (e.g., Gryspeerdt et al., 2021; Glassmeier et al., 2021) brought attention to the timescale for $L$ to adjust. Since

$$k' = \frac{\delta l'}{\delta n} - k\frac{\delta n'}{\delta n}, \tag{2}$$

where the apostrophe indicates the time derivative ($\mathrm{d}/\mathrm{d}t$), both the timescales for $L$ adjustment and $N$ evolution contribute to the timescale for $k$ (Gryspeerdt et al., 2022). Glassmeier et al. (2021) reported that $k$ approaches a "steady state" slope ($k_\infty$) around $-0.64$ with an adjustment timescale around 20 h based on the analysis of a large-eddy simulation (LES) ensemble of more than 100 nocturnal marine statocumuli. Other studies have reported shorter adjustment timescales (e.g., Rahu et al., 2022; Prabhakaran et al., 2023). Reconciling different estimates of adjustment timescales may require connecting them to

various other timescales previously discovered in the marine stratocumulus-topped BLs (STBLs), e.g., the inversion adjustment timescale (Schubert et al., 1979), the thermodynamic timescale (Schubert et al., 1979; Bretherton et al., 2010), and a short timescale associated with the strong feedback between entriainment velocity, $L$, and BL turbulence (Zhu et al., 2005; Bretherton and Blossey, 2014) as shown by Jones et al. (2014).

One less investigated aspect of the $L$ adjustment in the non-precipitating regime is its dependence on $N$. The effects of several aforementioned mechanisms for entrainment enhancement by increasing $N$ should saturate at high $N$ where a further increase in $N$ does not significantly affect the drop size, suggesting that the negative $k$ caused by these mechanisms at relatively low $N$ should become less negative at high $N$. Results from some previous studies (Lu and Seinfeld, 2005; Chen et al., 2011) support this expectation. Further narrowing down the uncertainty in the $N$-dependence of $k$ is valuable because it may reveal optimal MCB seeding strategies and provide an assessment of the integrated brightening effects as the seeded clouds lose $N$ through dilution (e.g., by mixing with FT air).

In this study, we use LES to investigate the magnitude and timescale of the $L$ adjustment for nocturnal non-precipitating marine stratocumuli. Even though applications like MCB intrinsically involve cloud evolution during the day, the nighttime evolution is still important because the high-$N$ conditions due to previous seeding may continue into the nighttime and the nighttime evolutions of the clouds and the BL shape the evolution over the course of the following day (Sandu et al., 2008; Chun et al., 2023; Zhang et al., 2024). This choice of scope also reduces the processes involved and simplifies the problem.

## 2 Method

### 2.1 Model and shared simulation configurations

LES simulations are performed using the System for Atmospheric Modeling (SAM; Khairoutdinov and Randall, 2003), version 6.10.10. The choices of specific schemes (e.g., numerical schemes for the dynamics, physical parameterizations, etc.) are identical to those described in the first two paragraphs of Section 2 in Chen et al. (2024).

Simulation configurations largely follow the configurations used in Glassmeier et al. (2021). The simulation domain is $48 \times 48 \times 2\,\mathrm{km}^3$ in the $x$, $y$, and $z$ dimensions with 200-m horizontal and 10-m vertical grid spacings. Periodic lateral boundary conditions and a damping layer from 1.5 km to domain top are used. The subsidence profile follows

$$w_\mathrm{s} = \begin{cases} -Dz,\ z < 1600\,\mathrm{m} \\ 0\,\mathrm{m\,s^{-1}},\ z \geq 1600\,\mathrm{m}, \end{cases} \tag{3}$$

where the divergence $D = 3.75 \times 10^{-6}\,\mathrm{s^{-1}}$ is based on the DYCOMS-II RF02 case (Wyant et al., 2007; Ackerman et al., 2009). The large-scale wind speed is zero at all levels so that there is no mean shear to drive turbulence. A constant surface aerosol flux of 70 $\mathrm{cm^{-2}\ s^{-1}}$ is prescribed to be consistent with Glassmeier et al. (2021). The time step is 1 s and the radiative scheme is called once every 10 s.

The main differences from Glassmeier et al. (2021) lie in other lower boundary conditions. We calculate surface fluxes interactively, instead of prescribing constant values based on DYCOMS-II RF02. A wind speed of 7 m $\mathrm{s^{-1}}$, based on ERA5

climatology (Hersbach et al., 2020), is added to the surface local wind fluctuation when calculating surface sensible and latent heat fluxes. See details in Section 2 and Appendix A in Chen et al. (2024). The sea surface temperature (SST) is case-dependent and fixed at 0.5 K warmer than the initial surface air temperature, instead of being fixed at the same value for all simulations.

### 2.2 Case definition and experiment design

We arbitrarily select 22 non-precipitating cases from Glassmeier et al. (2021). As shown in Figure S1, they span a wide range in the $N$–$L$ plane and the plane of $L$ and inversion base height ($z_\mathrm{i}$). Here and throughout the manuscript, $N$ is characterized by the mean cloud droplet number concentration for cloudy columns with hydrometeor optical depth greater than 1 and $L$ by the domain mean LWP, both following Glassmeier et al. (2021). In Glassmeier et al. (2021), a case is defined by its initial thermodynamic and aerosol profiles that are controlled by six parameters: initial boundary layer (BL) depth ($h_\mathrm{mix}$) randomly drawn from 500 to 1300 m, BL liquid water potential temperature ($\theta_\mathrm{l}$) from 284 to 294 K, BL total water mixing ratio ($q_\mathrm{t}$) from 6.5 to 10.5 g kg$^{-1}$, $\theta_\mathrm{l}$ jump across the inversion base ($\Delta\theta_\mathrm{l}$) from 6 to 10 K, $q_\mathrm{t}$ jump ($\Delta q_\mathrm{t}$) from $-10$ to $-6$ g kg$^{-1}$, and aerosol mixing ratio ($N_\mathrm{a}$) from 30 to 500 mg$^{-1}$. As shown in Figure S2, our 22 cases also cover the pair-wise spaces of the first five parameters, which are hereafter collectively referred to as the meteorological condition (MC). Note that the ensemble in Glassmeier et al. (2021) is characterized by a relatively dry free-troposphere (FT) with FT $q_\mathrm{t}$ between 0.2 and 2.8 g kg$^{-1}$ (see the BL $q_\mathrm{t}$–$\Delta q_\mathrm{t}$ panel in Figure S2). Our MCs are the same in this respect.

We use the MCs from these 22 cases to set up the initial $\theta_\mathrm{l}$ and $q_\mathrm{t}$ profiles but configure the aerosol as follows. For each MC, we first find a BASE run by setting the initial $N_\mathrm{a}$ throughout the domain to a low value and perform a 36-h simulation so that the trajectory of the simulation in the $N$–$L$ plane is close to the approximate threshold for precipitation (defined as the line that corresponds to a characteristic cloud-top mean drop radius of 12 $\mu$m) but the simulated cloud remains non-precipitating (defined as a cloud-base precipitation rate of less than 0.5 mm day$^{-1}$; Wood, 2012). Then, we seed the BASE run at 12 h after the beginning of the simulation by uniformly increasing the total number mixing ratio (i.e., the sum of aerosol, cloud droplet, and rain drop number mixing ratios) from the surface to 400 m for 30 min at constant rates estimated to achieve 7 $N$ targets: 95, 125, 165, 225, 300, 400, 550 cm$^{-3}$. A seeded run for a given MC is only performed if the $N$ target is greater than $N$ in the BASE run at the end of its 36-h simulation. The seeded runs are then simulated for 24 h. Due to various processes affecting $N$, the actual $N$ for seeded runs aimed at these 7 $N$ targets reach about 120, 135, 165, 200, 250, 310, and 400 cm$^{-3}$ at 36 h, still spanning a relatively wide range. We refer to seeded runs by their final $N$ as N120, N135, N165, N200, N250, N310, N400, respectively.

The simulation set described above is referred to as MAIN. Additional sensitivity runs are performed and will be introduced as needed.

115 ## 2.3 Budget analysis

### 2.3.1 $L$ budget

The $L$ budget is diagnosed based on mixed-layer theory (MLT; Lilly, 1968; Wood, 2007; van der Dussen et al., 2014; Ghonima et al., 2015; Hoffmann et al., 2020) because the BLs in all simulations are close to well-mixed and overcast (Figure S3). We start with

120 $$L = \frac{1}{2}\Gamma_{\rm l}\langle\rho_0\rangle(z_{\rm i}-z_{\rm cb})^2, \tag{4}$$

where $\Gamma_{\rm l}$ is the rate of change of liquid water mixing ratio ($q_{\rm l}$) with height in an adiabatic cloud, $\langle\rho_0\rangle$ is the cloud-layer mean air density, and $z_{\rm cb}$ is the domain mean cloud base. Then,

$$\begin{aligned} L' &= \frac{1}{2}\left(\Gamma_{\rm l}\langle\rho_0\rangle\right)'(z_{\rm i}-z_{\rm cb})^2 + \Gamma_{\rm l}\langle\rho_0\rangle(z_{\rm i}-z_{\rm cb})\left[z_{\rm i}'-z_{\rm cb}'\right] \\ &\approx \Gamma_{\rm l}\langle\rho_0\rangle(z_{\rm i}-z_{\rm cb})\left[z_{\rm i}'-z_{\rm cb}'\right] \\ &\approx \Gamma_{\rm l}\langle\rho_0\rangle(z_{\rm i}-z_{\rm cb})\left[z_{\rm i}'-\left(\frac{\partial z_{\rm cb}}{\partial\langle\theta_{\rm l}\rangle}\langle\theta_{\rm l}\rangle' + \frac{\partial z_{\rm cb}}{\partial\langle q_{\rm t}\rangle}\langle q_{\rm t}\rangle'\right)\right], \end{aligned} \tag{5}$$

where $\langle\theta_{\rm l}\rangle$ and $\langle q_{\rm t}\rangle$ are the BL mean $\theta_{\rm l}$ and $q_{\rm t}$, and $\partial z_{\rm cb}/\partial\langle\theta_{\rm l}\rangle$ and $\partial z_{\rm cb}/\partial\langle q_{\rm t}\rangle$ are the sensitivities of $z_{\rm cb}$ to BL warming/cooling and drying/moistening. Although $\Gamma_{\rm l}$, $\langle\rho_0\rangle$, $\partial z_{\rm cb}/\partial\langle\theta_{\rm l}\rangle$, and $\partial z_{\rm cb}/\partial\langle q_{\rm t}\rangle$ also change slightly with time, we use their time-dependent values in Eq. (5) but ignore terms containing their temporal rates of changes when expanding $L'$. The contributions of various physical processes to $L'$ are then diagnosed via their effects on $\langle\theta_{\rm l}\rangle'$, $\langle q_{\rm t}\rangle'$, and $z_{\rm i}'$:

130 $$L_P' = \Gamma_{\rm l}\langle\rho_0\rangle(z_{\rm i}-z_{\rm cb})\left[z_{{\rm i},P}'-\left(\frac{\partial z_{\rm cb}}{\partial\langle\theta_{\rm l}\rangle}\langle\theta_{\rm l}\rangle_P' + \frac{\partial z_{\rm cb}}{\partial\langle q_{\rm t}\rangle}\langle q_{\rm t}\rangle_P'\right)\right], \tag{6}$$

where tendency terms with a subscript $P$ indicate the contributions of $P$, which refers to one of cloud-top entrainment (ENTR), radiation (RAD), subsidence (SUBS), surface fluxes (SURF), and precipitation (PRCP).

Regarding $\langle\theta_{\rm l}\rangle'$ and $\langle q_{\rm t}\rangle'$, we express the contributions of a process in the form of the BL flux divergences of $\theta_{\rm l}$ and $q_{\rm t}$ ($\Delta F_{\langle\theta_{\rm l}\rangle,P}$ and $\Delta F_{\langle q_{\rm t}\rangle,P}$) evenly distributed over the entire BL depth, assuming that the BL is well-mixed,

135 $$\langle\theta_{\rm l}\rangle_P' = \frac{\Delta F_{\langle\theta_{\rm l}\rangle,P}}{\langle\rho_0\rangle_{\rm BL} z_{\rm i}},\ \langle q_{\rm t}\rangle' = \frac{\Delta F_{\langle q_{\rm t}\rangle,P}}{\langle\rho_0\rangle_{\rm BL} z_{\rm i}}. \tag{7}$$

For RAD, $\Delta F_{\langle\theta_{\rm l}\rangle,{\rm RAD}}$ is the radiative heating rate integrated from the surface to $z_{\rm i}$. For SURF, the corresponding fluxes (i.e., surface sensible and latent heat fluxes) are only defined at the surface but we still express its contributions in terms of flux divergences with the corresponding fluxes at $z_{\rm i}$ defined to be zero. Similarly, the flux divergences of PRCP are based on the surface precipitation rate and a zero precipitation flux at $z_{\rm i}$. SUBS is assumed to have no contributions to either $\langle\theta_{\rm l}\rangle'$ or $\langle q_{\rm t}\rangle'$ because the subsidence as prescribed in Eq. (3) only stretches the vertical profiles and does not move any air parcel across the $z_{\rm i}$. For ENTR, we simply attribute the differences between actual $\langle\theta_{\rm l}\rangle'$ and $\langle q_{\rm t}\rangle'$ and the sums of contributions of RAD, SURF, and PRCP to ENTR, closing the $\langle\theta_{\rm l}\rangle$ and $\langle q_{\rm t}\rangle$ budgets without residuals (see Section 4.1 in Chen et al., 2024 for details). We

still diagnose $\Delta F_{\langle\theta_\mathrm{l}\rangle,\mathrm{ENTR}}$ and $\Delta F_{\langle q_\mathrm{t}\rangle,\mathrm{ENTR}}$ from $\langle\theta_\mathrm{l}\rangle'_\mathrm{ENTR}$ and $\langle q_\mathrm{t}\rangle'_\mathrm{ENTR}$ following Eq. (7), which will be used later in the paper.

Regarding $z_\mathrm{i}'$, the SUBS term is simply the $w_\mathrm{s}(z_\mathrm{i})$ calculated using Eq. (3) and the ENTR term is the entrainment velocity

$$w_\mathrm{e} = z_\mathrm{i}' - w_\mathrm{s}(z_\mathrm{i}). \tag{8}$$

Finally, a residual term (RES) is calculated from the difference between the actual $L'$ and the sum of diagnosed terms to close the $L$ budget, even though it is not necessary for $\langle\theta_\mathrm{l}\rangle'$, $\langle q_\mathrm{t}\rangle'$, or $z_\mathrm{i}'$.

#### 2.3.2 $L$ budget in terms of BL flux divergence

We define

$$\mathcal{F}_P = \langle\rho_0\rangle_\mathrm{BL} z_\mathrm{i} z'_{\mathrm{i},P} - \left(\frac{\partial z_\mathrm{cb}}{\partial\langle q_\mathrm{t}\rangle}\Delta F_{\langle q_\mathrm{t}\rangle,P} + \frac{\partial z_\mathrm{cb}}{\partial\langle\theta_\mathrm{l}\rangle}\Delta F_{\langle\theta_\mathrm{l}\rangle,P}\right). \tag{9}$$

Obviously, the sum of $\mathcal{F}_P$ over ENTR, RAD, SUBS, SURF, and PREC is

$$\begin{aligned}\mathcal{F} = \sum_P \mathcal{F}_P &= \sum_P \left(\langle\rho_0\rangle_\mathrm{BL} z_\mathrm{i} z'_{\mathrm{i},P} - \left(\frac{\partial z_\mathrm{cb}}{\partial\langle q_\mathrm{t}\rangle}\Delta F_{\langle q_\mathrm{t}\rangle,P} + \frac{\partial z_\mathrm{cb}}{\partial\langle\theta_\mathrm{l}\rangle}\Delta F_{\langle\theta_\mathrm{l}\rangle,P}\right)\right)\\ &= \langle\rho_0\rangle_\mathrm{BL} z_\mathrm{i}\left[z_\mathrm{i}' - \left(\frac{\partial z_\mathrm{cb}}{\partial\langle\theta_\mathrm{l}\rangle}\langle\theta_\mathrm{l}\rangle' + \frac{\partial z_\mathrm{cb}}{\partial\langle q_\mathrm{t}\rangle}\langle q_\mathrm{t}\rangle'\right)\right].\end{aligned} \tag{10}$$

Compared with Eq. (5), $\mathcal{F}$ is essentially $z_\mathrm{i}' - z_\mathrm{cb}'$ multiplied by $\langle\rho_0\rangle_\mathrm{BL} z_\mathrm{i}$. Hereafter we refer to $\mathcal{F}$ as "the flux divergence for $L'$" to emphasize that we formulate it and its components by processes in terms of flux divergences $\Delta F_{\langle\theta_\mathrm{l}\rangle,P}$ and $\Delta F_{\langle q_\mathrm{t}\rangle,P}$.

Then, $L'$ can be written as

$$L' = p\zeta_\mathrm{c}\mathcal{F}, \tag{11}$$

where

$$p = \Gamma_\mathrm{l}\langle\rho_0\rangle / \langle\rho_0\rangle_\mathrm{BL} \tag{12}$$

is a prefactor and

$$\zeta_\mathrm{c} = 1 - z_\mathrm{cb}/z_\mathrm{i} \tag{13}$$

is the normalized cloud depth.

#### 2.3.3 $k$ evolution

Define

$$k_L' = \frac{\delta l'}{\delta n} = k\frac{\delta l'}{\delta l} \tag{14}$$

and

$$k'_N = -k\frac{\delta n'}{\delta n}. \tag{15}$$

Then, $k'$ is the sum of $k'_L$ and $k'_N$ (Eq. 2). And $k'_L$ can be further connected to the $L$ budget terms:

$$k'_L = \sum_P k'_P = \sum_P \frac{\delta l'_P}{\delta n} = \sum_P \frac{\delta\left(L'_P/L\right)}{\delta n}, \tag{16}$$

where $P$ refers to ENTR, RAD, SUBS, SURF, PRCP, and RES. In this paper, we do not further decompose $k'_N$.

## 3 Results

In this section, we show results based on the simulations described in Section 2.2 (the MAIN set). We first present an overview of our simulations, then show the $L$ and $k$ budgets, and end with results about timescales of $L$, $\delta L$, and $k$. In this section, all times refer to the time from the moment when seeding starts, $t_{\text{seeding}}$, which is 12 h from the beginning of the simulations.

### 3.1 Overview

We present an overview starting with the cloud states ($N$, $L$) and the $L$ adjustment slopes $k$ in two 1-h windows (Figure 1). Between 11 and 12 h, the 1-h median $L$ tends to be lower for higher $N$ for all MCs (Figure 1), consistent with the expected negative $L$ adjustment slope for non-precipitating stratocumuli.

The distributions of $k$ during the same time period are shown in Figure 1b. Since BASE, N165, N200, N250, N310, and N400 are always simulated for all 22 MCs, we divide the slopes between BASE and various seeded runs into six groups: BASE–N165, BASE–N200, BASE–N250, BASE–N310, BASE–N400, and BASE–other seeded runs. Each of the first five groups includes, obviously, 22 pairs of simulations, while the last one also includes 22 pairs (i.e., 4 BASE–N120 pairs and 18 BASE–N135 pairs). As shown with blue symbols in Figure 1b, the 1-h median $k$ between the simulation pairs becomes less negative for larger $N$. The medians for the six groups increase from $-0.38$ for BASE–other seeded runs, more negative than predicted by Glassmeier et al. (2021), to $-0.19$ for BASE–N400. This $N$-dependence of $k$ can also be seen in $k$ for all pairs of simulations with adjacent $N$, in other words, all line segments in Figure 1a. We divide these line segments into five groups: N165–N200, N200–N250, N250–N310, N310–N400, each including 22 line segments, and all remaining 44 line segments (i.e., 4 BASE–N120 segments, 14 BASE–N135 segments, 4 BASE–N165 segments, 4 N120–N135 segments, and 18 N135–N165 segments). The distributions of 1-h median $k$ for line segments in five groups are shown with red symbols in Figure 1b. The slopes are also less negative for pairs with larger $N$ with medians ranging from $-0.29$ to $-0.13$.

Between 23 and 24 h, the overall negative $L$ adjustment slope does not change much from between 11 and 12 h (cf. Figures 1c and 1a). The 1-h median $k$ between BASE and seeded runs (blue symbols) and for all line segments (red symbols) become more negative by about 0.06 (cf. Figures 1d and 1b), but $k$ values for most pairs are less negative than predicted by Glassmeier et al. (2021). The distributions of $k$ in all groups are broader than 12 h earlier, partially because $L$ for simulations within each MC fluctuate more (e.g., see polylines with red "×" and purple triangle in Figure 1c).

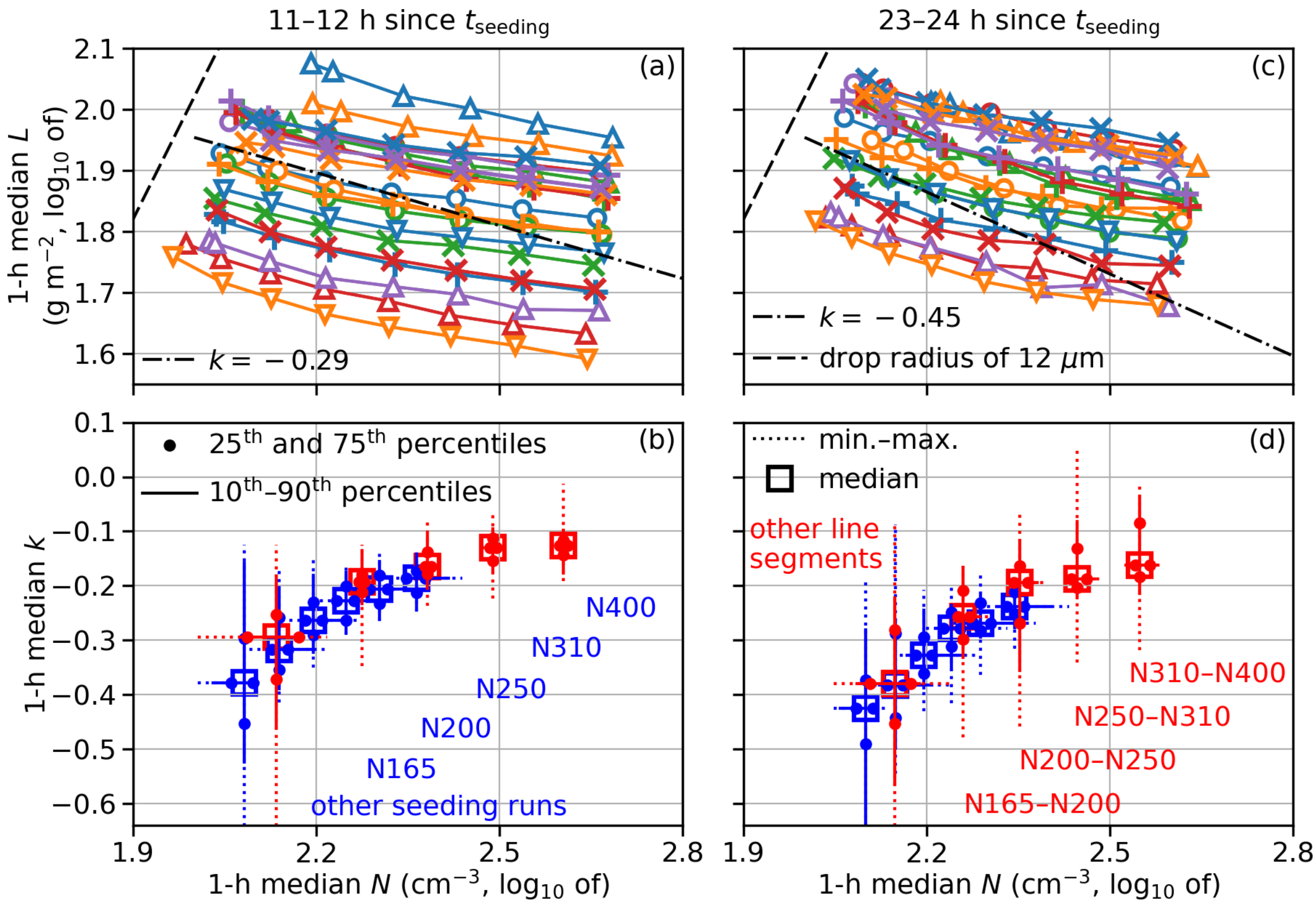

**Figure 1.** Overview of 1-h medians of cloud states ($N$, $L$) and $L$ adjustment slopes $k$ between 11 and 12 h (left column) and between 23 and 24 h (right column) in MAIN. Panels (a) and (c): Each curve connects simulations for one MC and there are at least 6 symbols along each curve (BASE, N165, N200, N250, N310, and N400). If present, symbols between BASE and N165 indicate N120 and N135. The combinations of the color and the shape of the symbol uniquely identify the 22 MCs, matching those in Figures S1 and S2. The dash-dotted black lines are reference lines showing $k$ at 12 h and 24 h assuming it exponentially converges to $-0.64$ with a timescale of 20 h, based on Glassmeier et al. (2021). The dashed black line is the precipitation line defined as the line that corresponds to a characteristic cloud-top mean drop radius of 12 $\mu$m. Panels (b) and (d): Distributions of 1-h median $k$ and 1-h median mid-point $N$ for various groups of simulation pairs. Blue symbols and lines are based on $k$ between BASE and seeded runs; red symbols are based on $k$ for line segments in Panels (a) and (c). See annotations in matching colors.

In all these results, $k$ is unlikely to reach $-0.4$ to fully compensate the Twomey effect.

Next, we present the time series of $L$, $N$, and $k$, focusing on the 22-MC composites (i.e., averages) for four aerosol configurations: BASE, N165, N250, and N400 (Figure 2). For BASE, $L$ increases from 71 g m$^{-2}$ to 92 g m$^{-2}$ from 0 h to 24 h since $t_{\text{seeding}}$. $N$ increases from 107 cm$^{-3}$ to 125 cm$^{-3}$, meaning the surface aerosol flux, which is the only aerosol source in BASE, dominates the net trend in $N'$. For seeded runs, $L$ and $N$ deviate from those for BASE. $L$ for N165 still increases, but at a slower rate; $L$ for N250 and N400 decreases first and then increases at even slower rates. $N$ peaks soon after 1 h, overshooting the $N$ target (165, 300, and 550 cm$^{-3}$ for N165, N250, and N400) by a few percent, then decreases relatively quickly for about

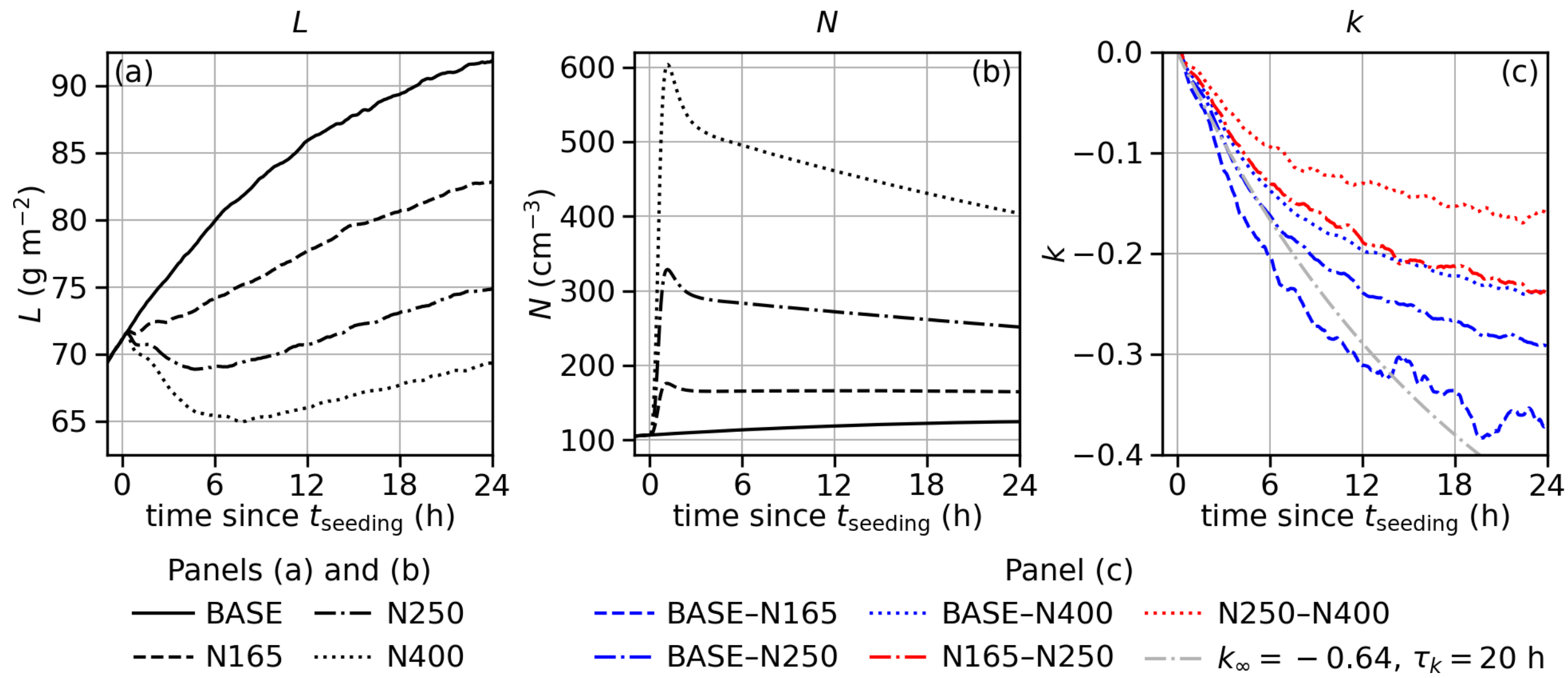

**Figure 2.** Time series of (a) $L$, (b) $N$, and (c) $k$ averaged across simulations for BASE, N165, N250, and N400.

2 h. This behavior is driven by the activation of the seeded aerosol and some initial adjustment of the boundary layer (BL) to the seeding. After about 5 h, $N$ decreases steadily towards the end of the simulations. For N400, this indicates the dominance of the dilution by the mixing between BL and FT air, given the negligible collision-coalescence at high $N$; for N165, multiple processes balance each other and $N$ does not change much. Both the $k$ between BASE and seeded runs and between seeded runs (blue and red curves in Figure 2c) become negative after the seeding starts, initially quickly and then more slowly. At 24 h, $L$, $N$, and $k$ for most aerosol configurations have not reached their steady states.

### 3.2 *L* budget

In this subsection, we examine the time series of the $L$ budget, again focusing on the 22-MC composites for BASE, N165, N250, and N400. Since the budget terms are noisier than $L$, the results are smoothed with a 4-h running average and plotted at the end of the 4-h window. Recall that time is relative to $t_{seeding}$, i.e., curves before 4 h are affected by results before the seeding, which are the same for BASE, N165, N250, and N400. Due to the smoothing, the data shown for the first few hours since $t_{seeding}$ are not suitable for directly estimating timescales, which will be investigated with unsmoothed data in Section 3.4, but the main features described in this subsection are valid.

We start with the time series of the flux divergence for $L'$ ($\mathcal{F}$) in Figure 3a. The variations in time and between BASE and seeded runs are much smaller than the range of values spanned by different terms. To see details, we also show all combinations of budget terms and aerosol configurations as the differences from the values at 0 h in Figures 3b and 3c. As expected, the contributions of ENTR and SUBS are negative (thinning cloud layer) and those of RAD and SURF are positive (deepening cloud layer). Since all simulated clouds are non-precipitating, the PRCP term is calculated for a precise RES term but omitted from the plot. The RES is close to 0, indicating a good closure of the budget (Figure 3b).

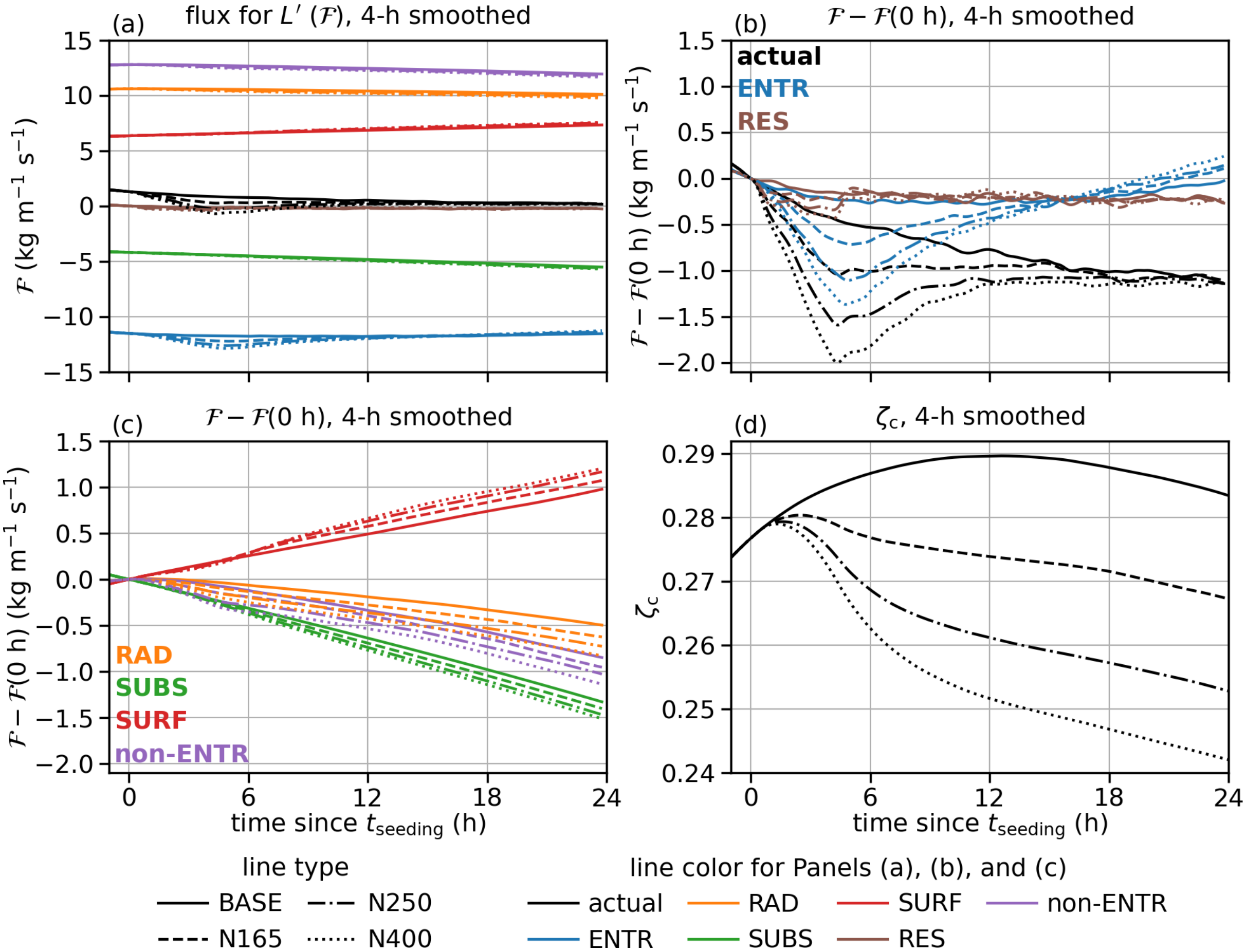

**Figure 3.** Four-hour smoothed time series of variables related to the flux divergence for $L'$ ($\mathcal{F}$) and the normalized cloud depth ($\zeta_c = 1 - z_{cb}/z_i$), averaged across simulations for BASE, N165, N250, and N400: (a) actual $\mathcal{F}$ and its budget terms, minus values at $t_{seeding}$, (b) actual $\mathcal{F}$ and its ENTR and RES terms, minus values at $t_{seeding}$, (c) RAD, SUBS, SURF, and the sum of these three non-ENTR terms for $\mathcal{F}$, and (d) $\zeta_c$.

For seeded runs, $\mathcal{F}$ for N400 becomes less positive than for BASE after the seeding starts, then turns negative, and then slowly returns to the $\mathcal{F}$ for BASE (Figure 3a). The response in $\mathcal{F}$ to seeding is dominated by the ENTR term (Figure 3b). This is consistent with our understanding that the entrainment is enhanced by an increase in $N$. The RAD term is slightly weaker (less positive) and the SURF and the SUBS terms are slightly stronger (more positive and more negative) in seeded runs, but the sum of all these three non-ENTR terms is slightly weaker (less positive). Even though the 4-h running average smooths the time series, it is clear that the response in SURF (Figure 3c) is delayed compared with the responses in ENTR and RAD. N165 and N250 show similar behavior with weaker deviation from BASE.

The normalized cloud depth $\zeta_c$ for seeded runs become less than BASE (Figure 3d), mainly because the cloud layers become thinner than BASE, and not because of the faster growth of $z_i$ due to enhanced entrainment (shown later).

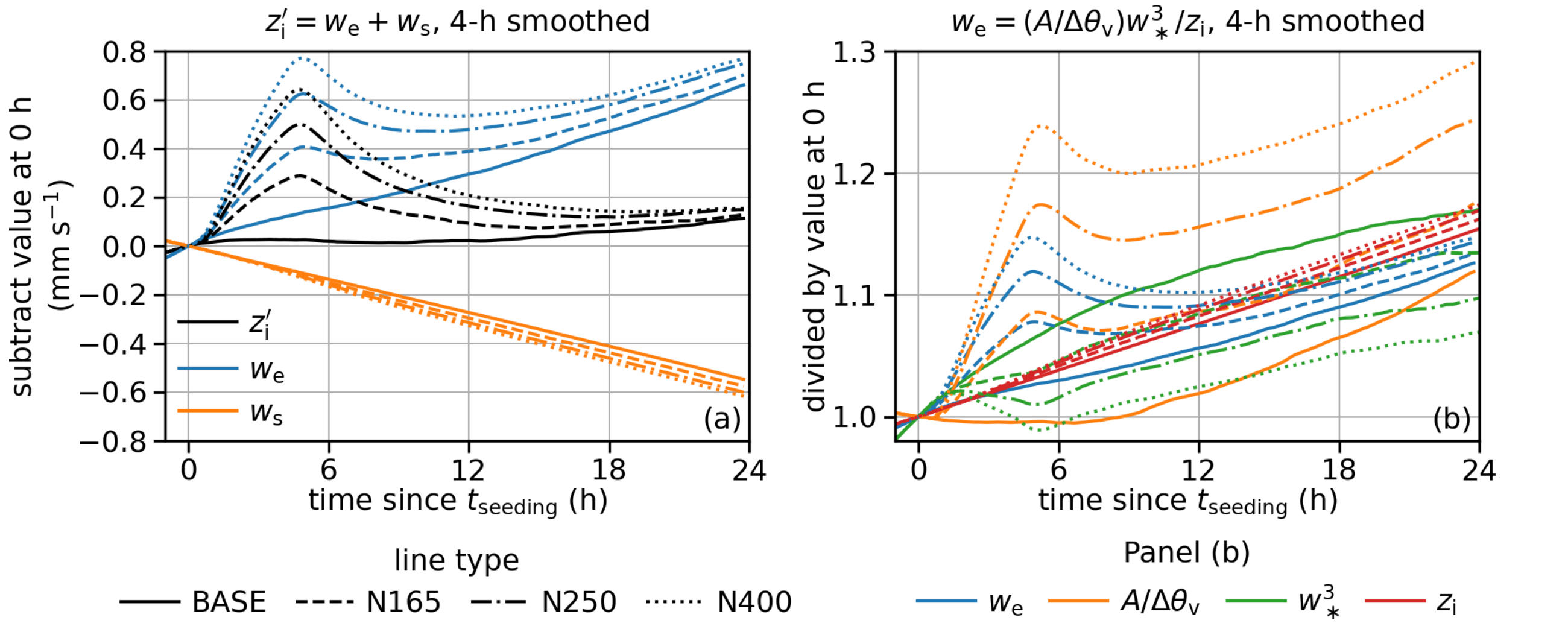

**Figure 4.** Same as Figure 3 but for variables related to the entrainment velocity ($w_e$): (a) $z_i'$, $w_e$, and $w_s$, minus values at $t_{seeding}$ and (b) components in $w_e$ parameterization $w_e = (A/\Delta\theta_v)w_*^3/z_i$, divided by values at $t_{seeding}$. See text around Eqs. (17) and (18) for definitions of symbols.

We take a closer look at time series relevant to the entrainment since it dominates the response in $\mathcal{F}$. The entrainment velocity $w_e$ dominates the response in $z_i'$ to seeding (Figure 4a). Initially, the seeded runs entrain much more strongly than BASE, causing faster growth of $z_i$; after a few hours, this difference starts to narrow. This behavior is similar to the results from Prabhakaran et al. (2023) (see their Figure 3g). Since the $\Delta\theta_l$ and $\Delta q_t$ in BASE only change slightly over time (Figure S4a) and the $\Delta\theta_l$ and $\Delta q_t$ respond weakly to seeding (Figure S4b), the behavior of $w_e$ dominates the response in $\mathcal{F}_{ENTR}$ to seeding. At the end of the simulation, $w_e$ in N400 is faster than in BASE by less than 0.2 mm s$^{-1}$, while the subsidence velocity $w_s$ is more negative by less than 0.1 mm s$^{-1}$. As a result, $z_i$ only grows marginally faster in seeded runs at 24 h since $t_{seeding}$.

We further break down $w_e$ following

$$w_e = \left(\frac{A}{\Delta\theta_v}\right)\frac{w_*^3}{z_i}, \tag{17}$$

where $A$ is an entrainment efficiency, $\Delta\theta_v$ is the virtual temperature jump across $z_i$, and $w_*$ is the convective velocity scale defined as

$$w_* = \left(\frac{2.5g}{T_0}\int_0^{z_i}\overline{w'\theta_v'}\mathrm{d}z\right)^{1/3}, \tag{18}$$

where $g$ is gravitational acceleration and $T_0$ is a temperature scale. We use $w_*^3$ and $z_i$ diagnosed by SAM and calculate $A/\Delta\theta_v$ as one term based on Eq. (17). The evolution of $A/\Delta\theta_v$ and its response to seeding is dominated by $A$, given the weak response in the $\Delta\theta_v$ inferred from the behavior of $\Delta\theta_l$ and $\Delta q_t$ (Figure S4). As shown in Figure 4b, $A/\Delta\theta_v$ in BASE increases over time as both $L$ and $N$ in BASE increase, consistent with previous works (Nicholls and Turton, 1986; Turton and Nicholls, 1987;

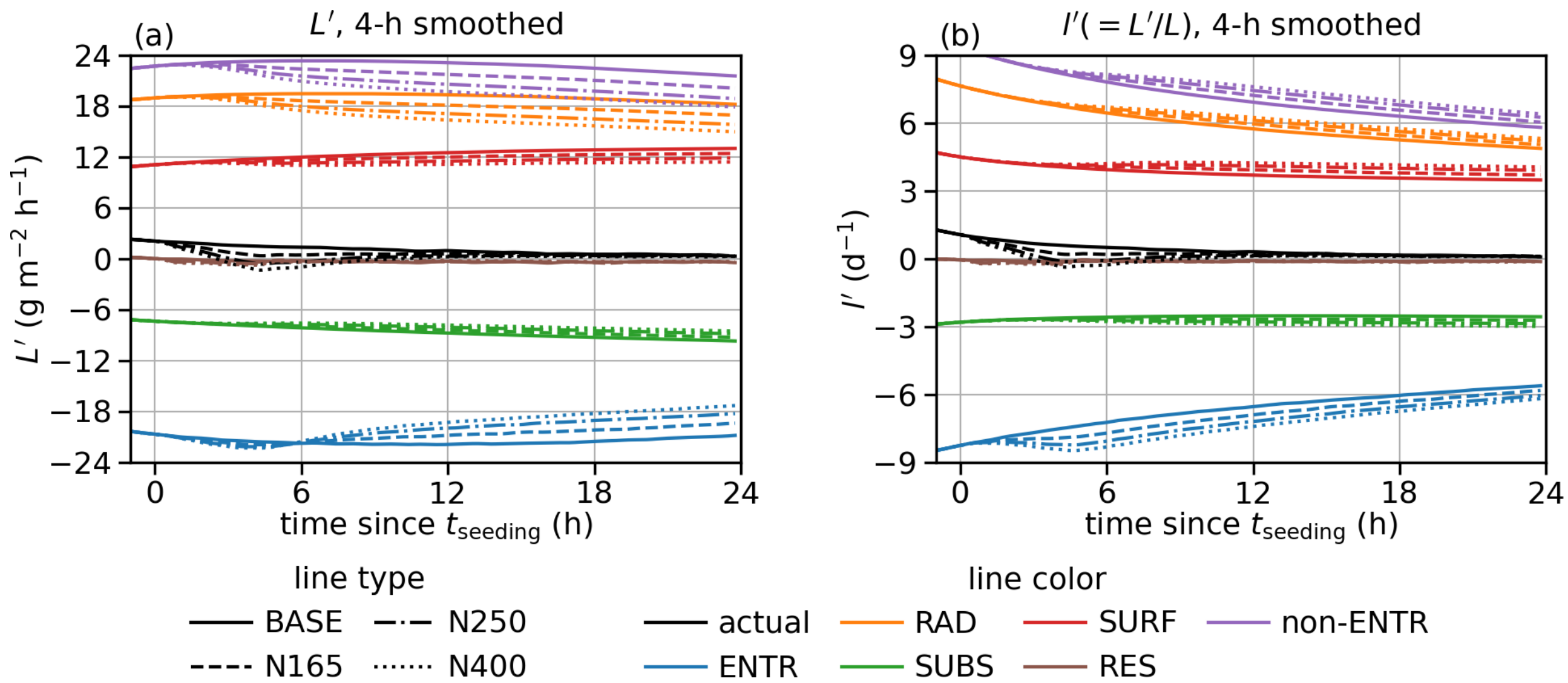

**Figure 5.** Same as Figure 3 but for (a) $L$ budget and (b) $l$ budget.

Bretherton et al., 2007); $A/\Delta\theta_{\mathrm{v}}$ in seeded runs increases with the amount of seeding, meaning the enhancement by increasing $N$ dominates over decreasing $L$. The evolution of $w_*^3$ in BASE and its response to seeding are consistent with the positive correlation between the buoyancy flux and $L$ (Bretherton and Wyant, 1997). The responses in $A/\Delta\theta_{\mathrm{v}}$ and $w_*^3$ to seeding are consistent with Chun et al. (2023). Different from $w_{\mathrm{e}}$, the magnitudes of the responses in both $A/\Delta\theta_{\mathrm{v}}$ and $w_*^3$ to seeding remain relatively large throughout the simulations.

Figure 5a shows the time series of $L$ budget terms. Since the prefactor $p$ in Eq. (11) evolves slowly in BASE and does not respond much to the seeding (not shown), the response in the $L$ budget to seeding is driven by $\mathcal{F}$ and $\zeta_{\mathrm{c}}$. The much more negative ENTR term for $\mathcal{F}$ in the seeded runs translates to a slightly more negative ENTR term for $L'$ (cf. Figures 3a and 5a). Later in the simulation, the contributions of ENTR, RAD, SUBS, and SURF are all weaker in the seeded runs due to smaller $\zeta_{\mathrm{c}}$. Figure 5b shows the time series of the $l$ budget terms. They are simply diagnosed as $l' = L'/L$ but still worth showing because $l'$ directly affects the evolution of $k$ (Eq. 14). With seeding, the contributions of all processes become stronger, opposite to the responses in the $L$ budget terms, because of the different $\zeta_{\mathrm{c}}$-dependence between $l'$ ($\propto \zeta_{\mathrm{c}}^{-1}\mathcal{F}$) and $L'$ ($\propto \zeta_{\mathrm{c}}\mathcal{F}$).

So summarize, entrainment dominates the response in $\mathcal{F}$ to seeding. The response in $\mathcal{F}_{\mathrm{ENTR}}$ is dominated by the response in $w_{\mathrm{e}}$, which peaks quickly and then decays towards the end of the simulation as a result of the balance between the persisting responses in $A$ and $w_*^3$. The responses in $\mathcal{F}$ and $\zeta_{\mathrm{c}}$ shape the evolution of $L'$ and $l'$.

### 3.3 $k$ budget

In this subsection, we examine the evolution of $k$ for two pairs of aerosol configurations: one between BASE and N165 and the other one between N250 and N400. The time series of $k$ budgets are even more noisy so we again apply the 4-h running average and plot smoothed results at the end of the 4-h window.

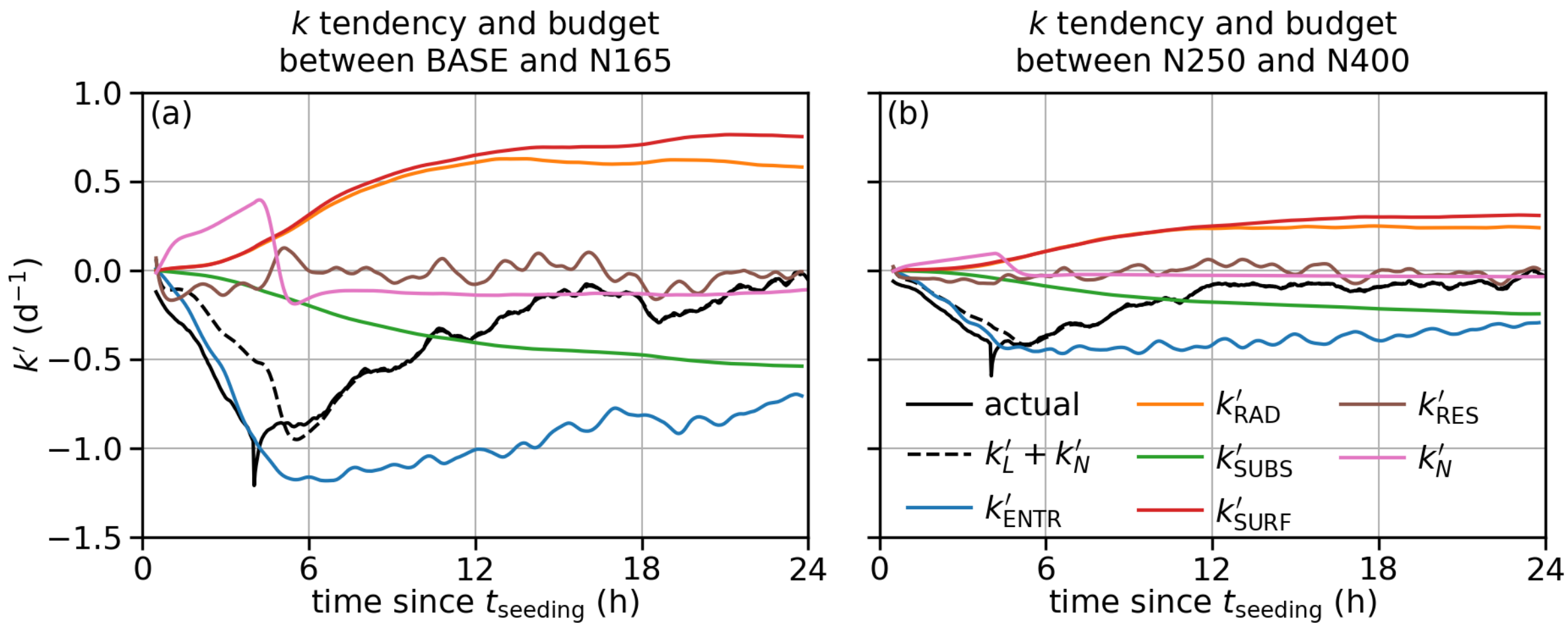

**Figure 6.** Four-hour smoothed time series for variables related to the $k$ budget, averaged across 22 MCs: (a) for $k$ between BASE and N165 and (b) between N250 and N400.

Figure 6a shows the budget for $k$ between BASE and N165. The actual $k'$ (from time series of $k$) becomes negative quickly after seeding starts, then decays towards 0, consistent with the evolution of $k$ for this pair (the dashed blue curve in Figure 2c). The $k'$ diagnosed as $k'_L + k'_N$ following Eq. (2) closely matches the actual $k'$ once we are beyond the first few hours since $t_{\mathrm{seeding}}$ (dashed black line in Figure 6a). ENTR and SUBS terms drive a more negative $k$, while the RAD and SURF impose a more positive trend. This qualitative behavior is expected from the $l$ budget in Figure 5b. Quantitatively, the ENTR term initially dominates, but later becomes more comparable in magnitude with RAD, SUBS, and SURF. Encouragingly, the RES term does not bias $k'$. The contribution of $N$ stays at a steady negative value after initial fluctuation because the $N$ in BASE and N165 get closer over time, making the negative $k$ even more negative (see Figure 2b and Eq. 15). In the last few hours of the simulation, $k'_N$ dominates the negative $k'$, even though the magnitude of $k'_N$ is small compared with the contributions of individual processes via $L'$. In other words, some of the decrease in $k$ between BASE and N165 in Figure 2c is due to the fact that BASE continues to gain $N$ but N165 does not (Figure 2b).

Figure 6b shows the budget for $k$ between N250 and N400. The qualitative behavior of all terms is similar to those for $k$ between BASE and N165 but with smaller magnitudes.

### 3.4 Timescales for $L$ and $\delta L$

Figure 7a shows the evolution of 22-MC composites of $L$ for BASE, N165, N250, and N400 in the phase space of $L$–$L'$. (Note that in this subsection, we use line colors, instead of line types, to indicate aerosol configurations.) The trajectories of all four aerosol configurations start from the circles near (71 g m$^{-2}$, 1.8 g m$^{-2}$ h$^{-1}$) at $t_{\mathrm{seeding}}$ and move towards the ends with no symbols. The circles along the trajectories are 4 h apart. Different from time series shown in Sections 3.2 and 3.3, these trajectories are based on the 2-min model output and not smoothed. For BASE, the trajectory is roughly linear with $L$

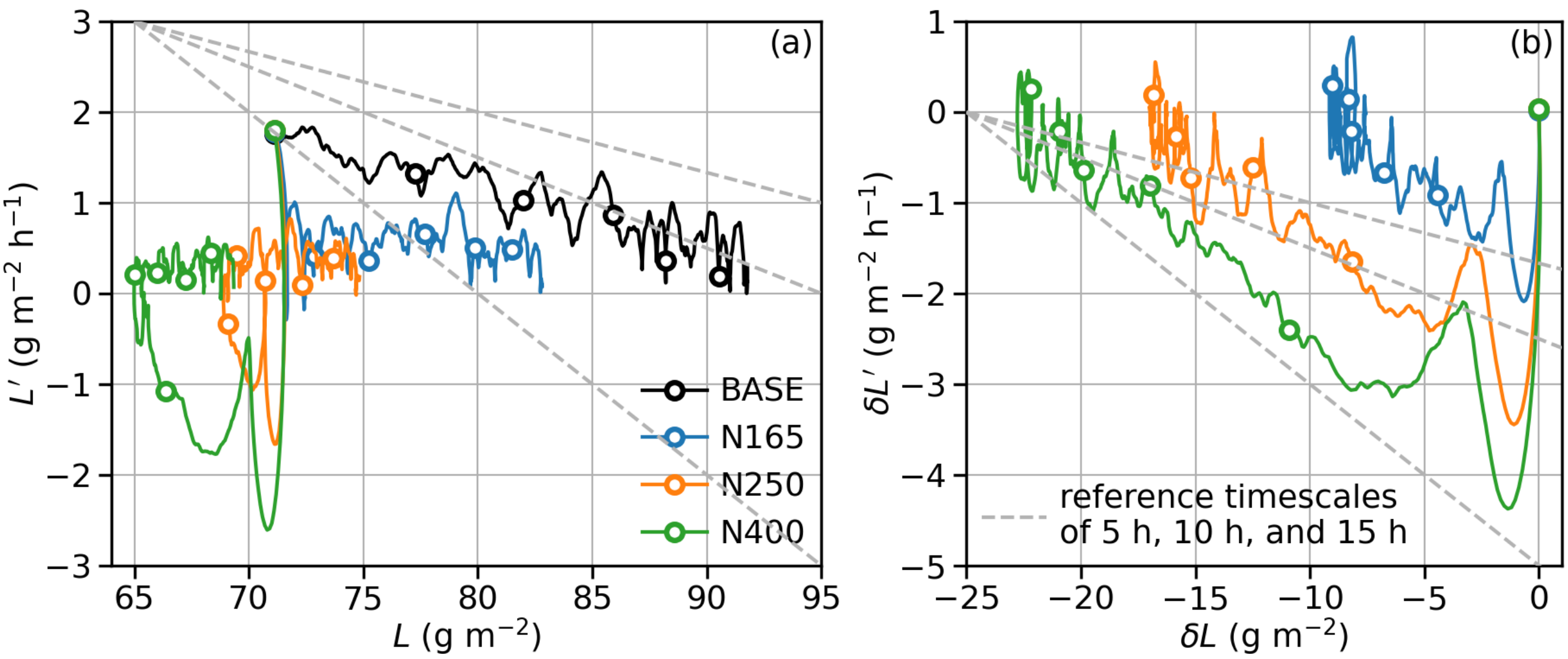

**Figure 7.** (a) Trajectories starting from $t_{\mathrm{seeding}} = 12\,\mathrm{h}$, averaged across simulations for BASE, N165, N250, and N400, in the $L$–$L'$ plane. All trajectories start around (71 g m$^{-2}$, 1.8 g m$^{-2}$ h$^{-1}$). Circles are 4 h apart from $t_{\mathrm{seeding}}$ to 20 h since $t_{\mathrm{seeding}}$. (b) Same as Panel (a) but in the $\delta L$–$\delta L'$ plane, where "$\delta$" indicates the deviation from BASE. All trajectories start around (0 g m$^{-2}$, 0 g m$^{-2}$ h$^{-1}$). For both panels, gray lines indicate the slopes for reference timescales of 5 h (the steepest one), 10 h, and 15 h (the least steep one).

increasing and $L'$ becoming less positive over time. Assuming a linear relation between $L'$ and $L$, a timescale $\tau$ can be defined as the inverse of the $L'$–$L$ slope:

$$L' = -(L - L_\infty)/\tau, \tag{19}$$

where $L_\infty$ is the value of $L$ for $L' = 0$. The solution to Eq. (19) is the equation for $L$ exponentially converging towards its steady state $L_\infty$ from initial state $L_0$ at an initial time $t_0$:

$$L(t) = L_\infty + (L_0 - L_\infty)\exp\left(-\frac{t - t_0}{\tau}\right). \tag{20}$$

(See the solid black curve in Figure 2a.) This evolution manifests in the $L$–$L'$ plane as a trajectory approaching ($L_\infty$, 0) and slowing down in that process. In our case, this timescale for BASE is between 10 and 15 h (cf. reference lines in Figure 7a).

For three seeded runs, the trajectories deviate from BASE dramatically in a few ways. For N400, the trajectory makes a loop. To be specific, $L'$ reduces quickly, becomes significantly negative, and then returns to a positive value. During the same time period, $L$ decreases to around 65 g m$^{-2}$. After 8 h since $t_{\mathrm{seeding}}$, $L'$ fluctuates around 0.3 g m$^{-2}$ h$^{-1}$ with $L$ increasing slowly. The trajectories for N165 and N250 behave similarly with smaller loops.

To understand this rather complicated evolution of seeded runs, we decompose their trajectories to the trajectory of BASE and a deviation from BASE, i.e.,

$$L = L_{\mathrm{BASE}} + \delta L,\ L' = L'_{\mathrm{BASE}} + \delta L', \tag{21}$$

where $\delta$ indicates the deviation from BASE. Figure 7b shows the trajectories of three seeded runs in the $\delta L$–$\delta L'$ plane. All three trajectories start from around (0 g m$^{-2}$, 0 g m$^{-2}$ h$^{-1}$) when seeding starts and again move towards ends with no symbols. For N400, $\delta L'$ quickly reaches its most negative point, after which it oscillates and then gradually becomes less negative towards 0. As a result, $\delta L$ becomes more negative but stabilizes. Considering that $L$ for BASE is approaching its steady state, this means that $L$ for N400 is probably approaching a different steady state. The trajectory approximately follows a line after 4 h since $t_{\mathrm{seeding}}$. Similarly, we can also define a timescale for this evolution of $\delta L$, $\tau_{\delta L}$, from the inverse of the $\delta L'$–$\delta L$ slope. A rough but reasonable estimate of $\tau_{\delta L}$ would be 5 h (cf. reference lines in Figure 7b).

To understand $\tau_{\delta L}$, we compare it with the timescales examined in Jones et al. (2014): a long timescale $\tau_3$ around 3 d, an intermediate timescale $\tau_2$ around 1 d, and a short timescale $\tau_1$ shorter than about 10 h (see their Table 2). Those authors attributed $\tau_3$ to the adjustment of $z_{\mathrm{i}}$ and $\tau_2$ to the thermodynamic adjustment in the BL. For $\tau_1$, they resorted to a mechanism called the entrainment–liquid flux (ELF) feedback, which was coined by Bretherton and Blossey (2014) to describe the mechanism for the response in both stratocumulus-topped BLs (STBLs) and cumulus-under-stratocumulus BLs to climate change. For well-mixed STBLs (i.e., conditions explored in our paper), it builds on the strong feedback between $w_{\mathrm{e}}$, $L$, and BL turbulence (Zhu et al., 2005): enhanced $w_{\mathrm{e}}$ warms and drys the BL, which reduces $L$, limits the buoyancy flux, weakens the BL turbulence, and eventually limits $w_{\mathrm{e}}$. Jones et al. (2014) showed that the signature of this feedback is the relatively steep $w_{\mathrm{e}}$–$\zeta_{\mathrm{c}}$ slope (see their Figures 2b and 3) which contributes to the short timescale in $L$ evolution via the cloud base height evolution (see their Eq. 24).

Our $\tau_{\delta L}$ is clearly comparable to this $\tau_1$. We examine the relation between the 22-MC composites of $\delta w_{\mathrm{e}}$ and $\delta\zeta_{\mathrm{c}}$ for three seeded runs in Figure 8, where $\delta$ indicates the deviation from BASE. All three ($\delta\zeta_{\mathrm{c}}$, $\delta w_{\mathrm{e}}$) trajectories start from (0, 0 mm s$^{-1}$) at $t_{\mathrm{seeding}}$. Using N400 as an example, $\delta w_{\mathrm{e}}$ increases and $\delta\zeta_{\mathrm{c}}$ decreases until the trajectory reaches its peak after about 2 h. (Note that the solid dots are 2 h apart.) Then, $\delta w_{\mathrm{e}}$ decreases towards 0 and $\delta\zeta_{\mathrm{c}}$ becomes more negative. The negative $\delta w_{\mathrm{e}}$–$\delta\zeta_{\mathrm{c}}$ slope beyond 2 h is considered "steep", compared with the values of around 15–30 mm s$^{-1}$ in Jones et al. (2014), suggesting a short timescale via the ELF feedback.

We further break down $\delta w_{\mathrm{e}}$ by linearizing Eq. (17):

$$\delta w_{\mathrm{e}} = \delta\left(\frac{A}{\Delta\theta_{\mathrm{v}}}\right)\frac{w_*^3}{z_{\mathrm{i}}} + \left(\frac{A}{\Delta\theta_{\mathrm{v}}}\right)\frac{\delta\left(w_*^3\right)}{z_{\mathrm{i}}} - \left(\frac{A}{\Delta\theta_{\mathrm{v}}}\right)\frac{w_*^3}{z_{\mathrm{i}}^2}\delta z_{\mathrm{i}}. \tag{22}$$

The three terms on the R.H.S. of this equation represent the impacts of responses in $A/\Delta\theta_{\mathrm{v}}$, $w_*^3$, and $z_{\mathrm{i}}$ in the seeded runs. This interpretation is valid because all factors other than the variables prefixed with "$\delta$" are based on the evolution of the BASE and do not correlate with the evolution of $\delta\zeta_{\mathrm{c}}$ (not shown). For N400, the $\delta\left(A/\Delta\theta_{\mathrm{v}}\right)$ term rapidly increases in the first 2 h after seeding starts (dashed green line). During the same time period, the $\delta\left(w_*^3\right)$ term first increases slightly, then starts to decrease as the enhanced entrainment warming/drying weakens the turbulence. Over the next 2 h, the $\delta\left(A/\Delta\theta_{\mathrm{v}}\right)$ decreases quickly, likely driven by the quick decrease in $N$ after peaking (Figure 2b). The trend in $\delta\left(w_*^3\right)$ is rather weak during this time period. Later, both $\delta\left(A/\Delta\theta_{\mathrm{v}}\right)$ and $\delta\left(w_*^3\right)$ terms contribute to the decrease in $\delta w_{\mathrm{e}}$. The $\delta z_{\mathrm{i}}$ term is negative all the time but its magnitude is small. The linearized equation does not fully recover the actual $\delta w_{\mathrm{e}}$ for N400 (see the gap between the green thick curve and the green thin curve with dots), which is not surprising given the relatively large change in $N$ between BASE and N400. However, the $\delta w_{\mathrm{e}}$–$\delta\zeta_{\mathrm{c}}$ slope is well captured. N165 and N250 behave qualitatively similar to N400.

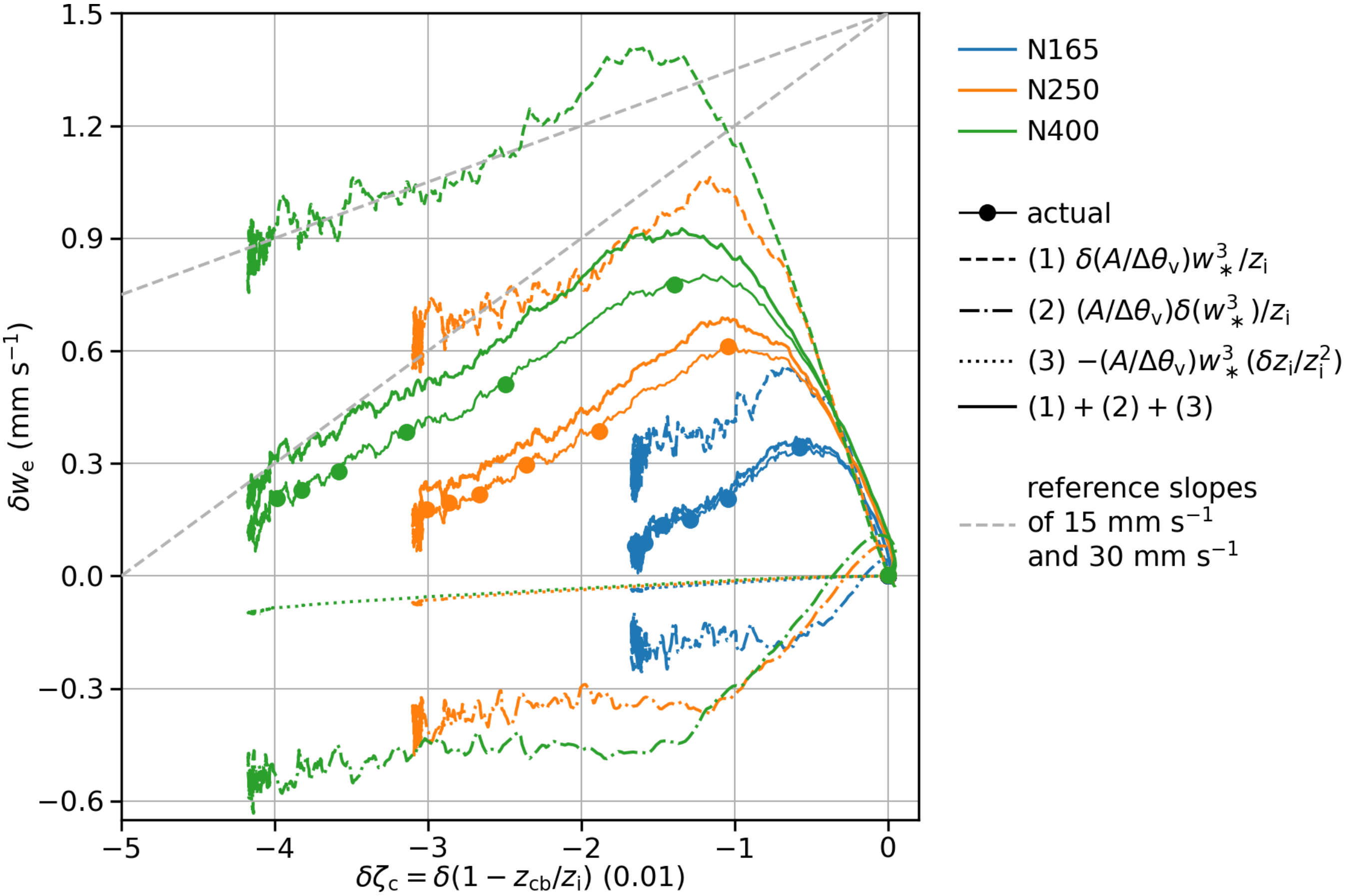

**Figure 8.** Same as Figure 7 but in the $\delta\zeta_c$–$\delta w_e$ plane. All trajectories start from (0, 0 mm s$^{-1}$) at $t_{\text{seeding}} = 12$ h. Dots are 2 h apart from $t_{\text{seeding}}$ to 12 h since $t_{\text{seeding}}$.

335 To summrize, by decomposing the $L$ in seeded runs into the $L_{\text{BASE}}$ and the deviation from it ($\delta L$, which is the $L$ adjustment), we show that $\delta L$ becomes negative following an exponential convergence. This evolution is governed by a short timescale of 5 h and lasts for 8–12 h. This short timescale is comparable with the one examined in Jones et al. (2014) which the authors attributed to the feedback between $w_e$, $L$, and BL turbulence (Zhu et al., 2005). We check the $\delta w_e$–$\delta\zeta_c$ slope in our simulations to conclude that our fast evolution is likely due to the same mechanism.

## 340 3.5 Timescales for $k$

Since the $k'$ time series is very noisy, the $k$ timescale is not readily identifiable from trajectories in the $k$–$k'$ plane. However, from Eqs. (2), (14), and (15) and results in Figure 6, one may anticipate that: when $k'_L$ dominates, the timescale for $k$ is close to the timescale for $l$, which is close to $\tau_{\delta L}$ when the range of $L$ variation discussed here is not too large; later when $k'_N$ becomes important the timescale for $k$ becomes longer because $k'$ is roughly constant and does not scale with $k$ anymore. To

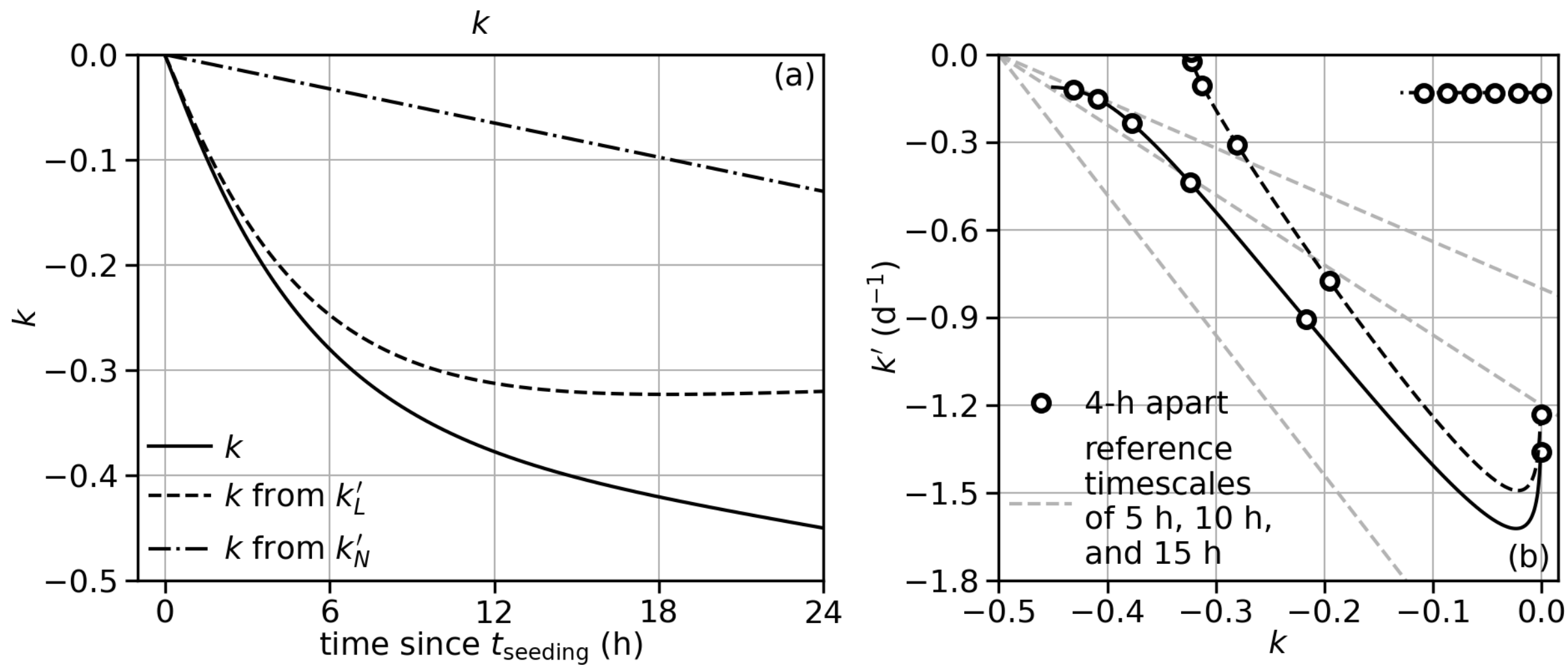

**Figure 9.** Evolution of $k$ from the ordinary differential equation (ODE) set for $k$ and $n$ based on Eqs. (2) and (15) with parameters representing the evolution of $k$ between BASE and N165: (a) time series of $k$ and the components from $k'_L$ and $k'_N$ and (b) trajectories of $k$ and its components in the $k$–$k'$ plane.

demonstrate this idea, we build an ordinary differential equation (ODE) set for $k$ (Eq. 2) and $n$ (rearranged from Eq. 15). To close these two ODEs, we assume that (1) the evolution of $L$ for BASE is entirely governed by Eqs. (19) and (20) (i.e., with no $k$ or $\delta n$ in these two equations), (2) $\delta L$ also exponentially converges (see Figure 7b), and (3) $k'_N$ is constant. With parameters representing the evolution of $k$ between BASE and N165 in our simulation, the results confirm the reasoning above (Figure 9).

## 4 Discussion

In this section, we first discuss the timescales associated with $L$ and $\delta L$ in our simulations with the timescales discovered in previous works. This helps us understand where these timescales fit into a bigger conceptual picture in both the model and the real world. Then we will discuss a few specific issues related to the $\delta L$, connecting back to this conceptual picture when necessary.

### 4.1 Comparison of $\delta L$ timescales with previous studies

Recall that in Bretherton et al. (2010) and Jones et al. (2014), the short timescale emerges after some perturbation is introduced into the mixed-layer model (MLM) and LES of stratocumulus. In Bretherton et al. (2010), this perturbation was unintentional: it was due to the inevitable mismatch between the initial states and the steady states of simulated clouds. It decays in one to two days as the system moves into the slow manifolds governed first by the thermodynamic timescale and later by the inversion adjustment timescale. In Jones et al. (2014), this perturbation was intentionally introduced to steady states in the model following van Driel and Jonker (2011). Their short timescale dominates for at least 12 h and decays later (e.g., their

Figure 3; also note that several diagnostics used to characterize the short timescale were based on the first 12-h results after perturbation).

In our simulations, the timescale for $L$ in BASE is about 15 h between $t_{\text{seeding}} = 12\,\text{h}$ and the end of the 36-h simulations (Figure 7a), suggesting that the BASE, after initial spin-up of turbulence, is moving towards the stage dominated by the thermodynamic timescale. The seeding introduces a perturbation into BASE during this transition. This is different from Jones et al. (2014), where the reference states are steady. Soon after the 30-min seeding ends, $\delta L$ evolves following a short timescale (Figure 7b) with signatures of the ELF feedback (Figure 8). Together with the evolution of $\delta N$, the $k$ evolution shows varying timescales even before the end of the simulations (Figure 9).

With this conceptual picture, we can anticipate the timescales found in our simulations to be transient. Even though it appears that both BASE and seeded runs are approaching some steady states within the 36-h simulations, it takes more than 10 days for them to eventually reach the steady states shown in Bretherton et al. (2010); the evolution will slow down during this period as the dominant timescale becomes longer due to the shift in the primary physical mechanism that drives the evolution. Hence, for both theoretical understanding and real-world application, it is important to know how long the short timescale dominates. From the composites based on MAIN, $\delta L$ and hence $k$ evolve according to the short timescale for 8–12 h.

### 4.2 Similarity of $\delta L$ evolution

In MAIN, even though the magnitude of $\delta L$ depends on $N$, the timescale of $\delta L$ is similar among N165, N250, and N400 (Figure 7b). As a result, the trajectories in the $\delta L$–$\delta L'$ plane are similar after appropriate scaling. To illustrate this feature, we divide $\delta L$ by $\delta L^*$, defined as the magnitude of the median $\delta L$ between 23 and 24 h since $t_{\text{seeding}} = 12\,\text{h}$; we divide $\delta L'$ by $\delta L'_*$, defined as $\delta L^*/5\,\text{h}$. With this scaling, a slope of $-1$ in the $\delta L/\delta L^*$–$\delta L'/\delta L'_*$ plane conveniently indicates a timescale of 5 h. The scaled trajectories of N165, N250, and N400 approximately collapse (Figure 10a). This is also true for similarly scaled $\delta l$–$\delta l'$ (Figure 10b) and scaled $\delta\zeta_\text{c}$ and scaled $\delta w_\text{e}$ (Figure 10c). (The scaling factor $\delta\zeta_\text{c}^*$ is defined as the magnitude of the median $\delta\zeta_\text{c}$ between 23 and 24 h since $t_{\text{seeding}}$; $\delta w_\text{e}^*$ equals to 20 mm s$^{-1} \cdot \delta\zeta_\text{c}^*$. With this scaling, a slope of 1 in the $\delta\zeta_\text{c}/\delta\zeta_\text{c}^*$–$\delta w_\text{e}/\delta w'_\text{e}$ plane indicates a $\delta w_\text{e}$–$\delta\zeta_\text{c}$ slope of 20 mm s$^{-1}$.)

To further understand this similar $\tau_{\delta L}$, we derive a simple model to connect $\delta w_\text{e}$ and $\delta\zeta_\text{c}$ with simplifications based on the phenomenology in MAIN. We start with the definition of $\zeta_\text{c}$ in Eq. (13). Assuming $z_\text{i}$ does not change with time (termed as "fixed boundary layer depth limit" in Jones et al., 2014),

$$\zeta'_\text{c} = \frac{z'_\text{i} - z'_\text{cb}}{z_\text{i}} = \frac{\mathcal{F}}{\langle\rho_0\rangle_\text{BL} z_\text{i}^2}. \tag{23}$$

Next, we decompose terms in Eq. (23) into a reference state (BASE in our case) and a deviation therefrom due to the seeding (again, prefixed with "$\delta$"). Considering that only ENTR flux divergences respond strongly to seeding (Figure 3),

$$\delta\zeta'_\text{c} = \frac{\delta\mathcal{F}}{\langle\rho_0\rangle_\text{BL} z_\text{i}^2} \approx \frac{\delta\mathcal{F}_\text{ENTR}}{\langle\rho_0\rangle_\text{BL} z_\text{i}^2} = -\frac{1}{\langle\rho_0\rangle_\text{BL} z_\text{i}^2}\left(\frac{\partial z_\text{cb}}{\partial\langle q_\text{t}\rangle}\delta\left(\Delta F_{\langle\theta_\text{l}\rangle,\text{ENTR}}\right) + \frac{\partial z_\text{cb}}{\partial\langle\theta_\text{l}\rangle}\delta\left(\Delta F_{\langle q_\text{t}\rangle,\text{ENTR}}\right)\right). \tag{24}$$

Again, due to the weak responses in $\Delta\theta_\text{l}$ and $\Delta q_\text{t}$, we further have

$$\delta\zeta'_\text{c} \approx -\frac{1}{\langle\rho_0\rangle_\text{BL} z_\text{i}^2}\left(\frac{\partial z_\text{cb}}{\partial\langle q_\text{t}\rangle}\Delta q_\text{t} + \frac{\partial z_\text{cb}}{\partial\langle\theta_\text{l}\rangle}\Delta\theta_\text{l}\right)\delta w_\text{e} = c_1\delta w_\text{e}, \tag{25}$$

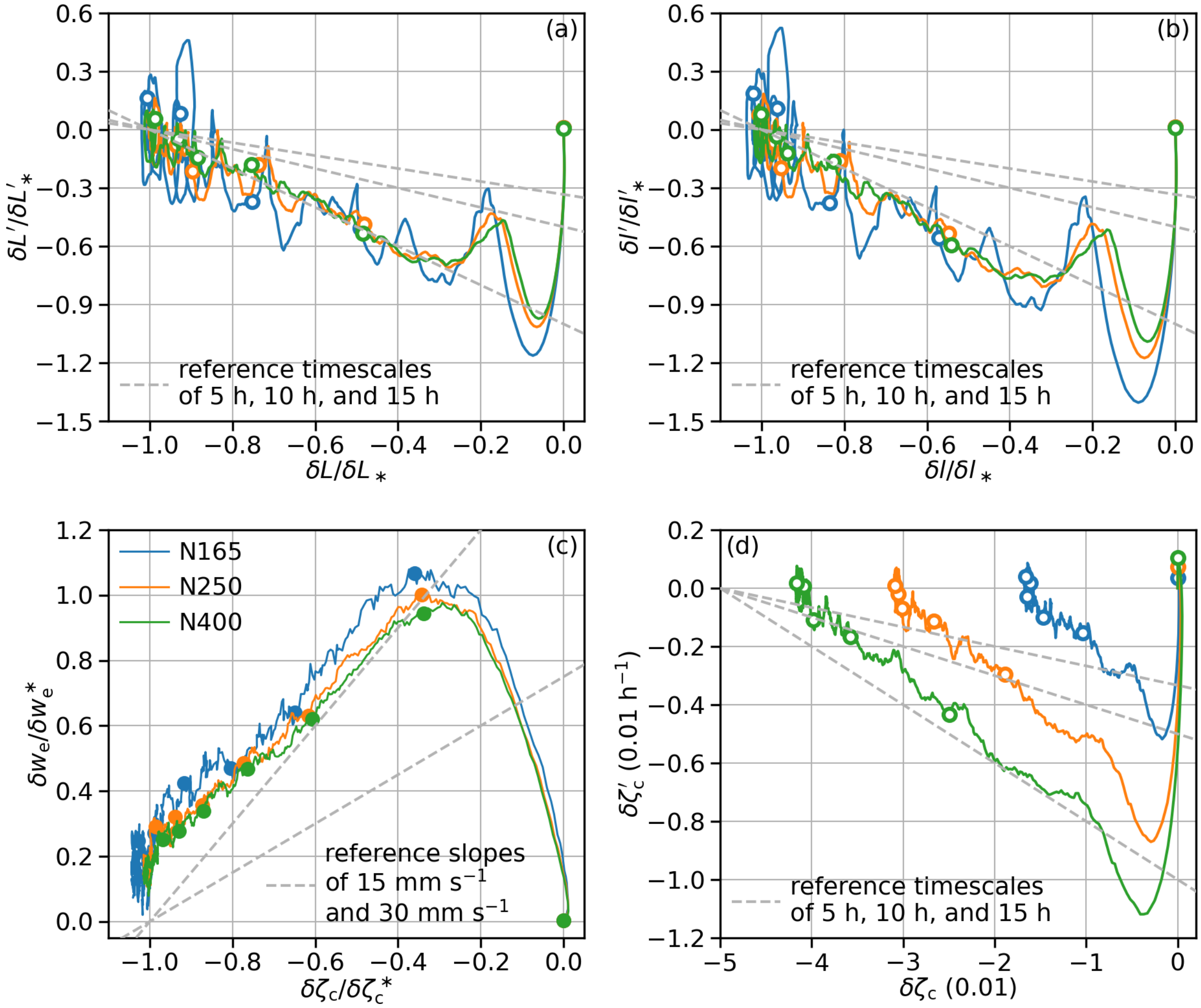

**Figure 10.** (a) Same as Figure 7b but for scaled ($\delta L$, $\delta L'$). (b) Same as Panel (a) but for scaled ($\delta l$, $\delta l'$). (c) Same as Figure 8 but for scaled ($\delta\zeta_c$, $\delta w_e$). (d) Trajectories averaged across 22 MCs, in the $\delta\zeta_c$–$\delta\zeta_c'$ plane; all trajectories start around (0, 0 h$^{-1}$) at $t_{seeding} = 12$ h. Circles are 4 h apart from $t_{seeding}$ to 20 h since $t_{seeding}$; solid dots are 2 h apart from $t_{seeding}$ to 12 h since $t_{seeding}$.

where we denote the prefactor in front of $\delta w_e$ with "$c_1$" to simplify the notation. Lastly, we approximate the trajectories of ($\delta\zeta_c$, $\delta w_e$) after 2 h since $t_{seeding}$ in Figure 8 with a linear function,

395 $$\delta w_e = c(\zeta_c - \delta\zeta_{c,\infty}) = \delta w_0 + c\delta\zeta_c, \quad (26)$$

where $c$ refers to the $\delta w_e$–$\delta\zeta_c$ slope, following the notation in Eq. (22) in Jones et al. (2014), and $\delta\zeta_{c,\infty}$ and $\delta w_0$ are parameters for this linear relation that can be interpreted as the steady state of $\delta\zeta_c$ and the initial $\delta w_e$. Substituting Eq. (26) into Eq. (25) reveals a linear relation between $\delta\zeta_c'$ and $\delta\zeta_c$

$$\delta\zeta_c' = c_1 c(\delta\zeta_c - \delta\zeta_{c,\infty})\,. \quad (27)$$

This is indeed the case with $\delta\zeta_c'$ and $\delta\zeta_c$ from the simulations (Figure 10d). Again, this means $\delta\zeta_c$ exponentially converges to its steady state $\delta\zeta_{c,\infty}$, governed by a timescale $1/(c_1c)$.

In Eq. (27), $c_1$ is mainly controlled by the meteorological conditions (see Eq. 25) in BASE; $c$ describes the phenomenology of the relation between $\delta w_e$ and $\delta\zeta_c$ in our simulations. In a MLM framework, one would be able to show that $c$ is the sum of terms controlled by both meteorology and aerosol (through the entrainment efficiency), e.g., by linearizing $w_e$ in Eq. (15) in Dal Gesso et al. (2014). This exercise is beyond the scope of current work.

Here, we make three points. First, even though the similarity among seeded runs is striking, it is more likely approximate than exact. Based on the conclusions from perturbing the linearized MLM in Jones et al. (2014), the relative changes in the short timescale are expected to directly connect to the relative changes in the entrainment efficiency, which is within 15% in MAIN (Figure 4b). With the noisy LES data, this difference may not be evident. If this is the case, the results suggest that the short timescale is mostly determined by the reference state, and that the difference between aerosol perturbations within a certain range only slightly modifies it. Second, the results in Jones et al. (2014) suggested that the similarity in $c$ also exists when perturbations are applied to state variables for steady states in MLM (see the similar $\delta w_e$–$\delta\zeta_c$ slopes in their Figure 3). For these perturbations, our simplifications (e.g., weak responses in the flux divergences by processes other than entrainment, jumps, and other factors) may not be applicable. It would be interesting to connect the response due to seeding to the response to these perturbations. Both points direct future work to addressing a broader question: how does the short timescale depend on meteorological and aerosol reference states and meteorological and aerosol perturbations? Lastly, as mentioned in Section 3.4, $\delta L$ starts the exponential convergence soon after the seeding stops. After that, the evolution of $\delta L$ is only controlled by two parameters: the initial $\delta L'$ (approximated by the most negative $\delta L'$ along each trajectory in Figure 7b) and the timescale $\tau_{\delta L}$. The similarity suggests that $\tau_{\delta L}$ is insensitive to the amount of seeded aerosol. Then the evolution of $\delta L$ (at least for the first 8–12 h) is controlled by a single parameter: the initial $\delta L'$, which is more negative for greater seeding amount.

### 4.3 A few sensitivity tests

We present five sensitivity sets in this section.

First, we argue that $k$ is more negative for low $N$ and less negative for high $N$. However, this statement is ambiguous because it takes two cloud states with different $N$ to define a $k$ and it is unclear whether we are talking about the background $N$ or seeded $N$ or some $N$ range that these two states span. Note that in MAIN, all seeded runs are based on the same BASE for each MC. Here, we modify the initial $N_a$ in the 22 BASE runs in MAIN so that their $N$ at 36 h of the simulations are close to 165 cm$^{-3}$. Then we seed these new BASE runs to reach $N$ around 250. We refer to the new BASE as B165 and the new seeded runs as B165N250. The evolution of $k$ between B165 and B165N250 averaged across MCs is quite similar to that between N165 and N250. Based on this results, we can generally claim that $k$ is more negative for lower background $N$.

The next three sensitivity runs show the impacts of three factors: $N$ evolution, non-steady BASE, and drifting FT $\theta_l$.

The evolving $N$ not only directly contributes to $k$ evolution through $k_N'$, but also has a footprint in $L'$ (e.g., Gryspeerdt et al., 2022). To be specfic, $L$ at a given time is the sum of initial $L$ and the integral of $L'$ during its evolution. For a pair simulations that reach the same $N$ through different evolutions, their $L$ may carry the "memory" of different $N$. In MAIN, $N$ evolves due

to surface flux of aerosol, dilution of BL aerosol through mixing with FT air, microphysical processes, and so on. Even though all these processes exist in nature, we perform simulations with approximately constant $N$ to quantify the impacts of the $N$ evolution. We seed all simulations including BASE in MAIN but (1) specify different initial seeded rates and (2) nudge total number concentrations in the BL after initial seeding finishes so that $N$ for all simulations stays about the same as the $N$ at the end of corresponding simulations in MAIN. In these simulations, the $k$ is less negative by about 0.05 for pairs between N165 and simulations with less $N$; the impacts on pairs with larger $N$ are less significant.

Our BASE in MAIN is not in steady state and it is unclear what the results would be if we were to let BASE evolve longer before seeding. We therefore re-do all seeded runs but delay the seeding time by 12 h to 24 h from the beginning of simulation. The $k$ between 11 and 12 h after this new seeding time are comparable to the previous $k$ around the same period of time after the original $t_{\mathrm{seeding}}$ (Figures 1a and 1c), pointing to the robustness of our results.

In MAIN, there is no nudging or tuning of subsidence to compensate the radiative cooling in FT. As a result, the FT air could cool down by about 0.5 to 1 K in 36-h simulations. This is not very large compared with our range of initial $\Delta\theta_{\mathrm{l}}$ from 6 to 10 K. Still, we want to make sure that this cooling does not introduce unintended $N$-dependence in simulation results. We therefore re-run the whole ensemble with FT $\theta_{\mathrm{l}}$ profiles from 100-m above the $z_{\mathrm{i}}$ and higher nudged to their initial values with a 30-min timescale. This nudging produces simulations with higher $L$ and lower $z_{\mathrm{i}}$, probably because it suppresses the cloud-top entrainment, but the impacts on $k$ are not evident.

To summarize, the sensitivity of results to (the exclusion of) $N$ evolution is consistent with our physical understanding; the sensitivity to seeding time and nudging of FT $\theta_{\mathrm{l}}$ profiles show the robustness of our results. Also, in all three sensitivity sets, $\tau_{\delta l}$ is shorter than 10 h. This insensitivity is probably because simulations in these sensitivity sets are close to well-mixed and overcast with similar dominant processes for BL turbulence (e.g., with no shear), as in MAIN.

Finally, we test the sensitivity of results to horizontal grid spacing. In MAIN and the three sensitivity sets, we use 200-m horizontal grid spacing, which is relatively coarse. We randomly pick one MC and repeat BASE, N165, N250, and N400 with two finer resolutions: (1) 100-m horizontal grid spacing and (2) 50-m horizontal grid spacing but reduce the horizontal domain size from 48 km to 24 km to save computational resources. For both resolutions, the vertical grid spacing stays unchanged at 10-m. The adjustment slopes produced by both fine resolution configurations are still more/less negative for lower/higher $N$ but the magnitudes are about 70–80 % of those in MAIN. The timescales for $\delta l$ are too noisy to precisely quantify from a single MC, but they are comparable to that in MAIN.

### 4.4 Impacts of perturbing initial $N_{\mathrm{a}}$ and fixed surface flux

Key differences between the design of MAIN in the current work and that of the LES ensemble set used in Glassmeier et al. (2021) (this set hereafter referred to as G21) include: (1) interactive surface fluxes in MAIN versus fixed surface fluxes in G21, (2) SST 0.5 K warmer than the initial surface air temperature in MAIN versus fixed SST in G21, and (3) seeding at 12 h to achieve regularly-spaced $N$ levels in MAIN versus randomized initial $N_{\mathrm{a}}$ in G21. We simulate one intermediate set (hereafter denoted with "INT" for "intermediate") for the 22 MCs used in MAIN to illustrate a few points. This set is configured with fixed surface sensible and latent heat fluxes and fixed SST that are identical to G21 and the aerosol configuration that is closer

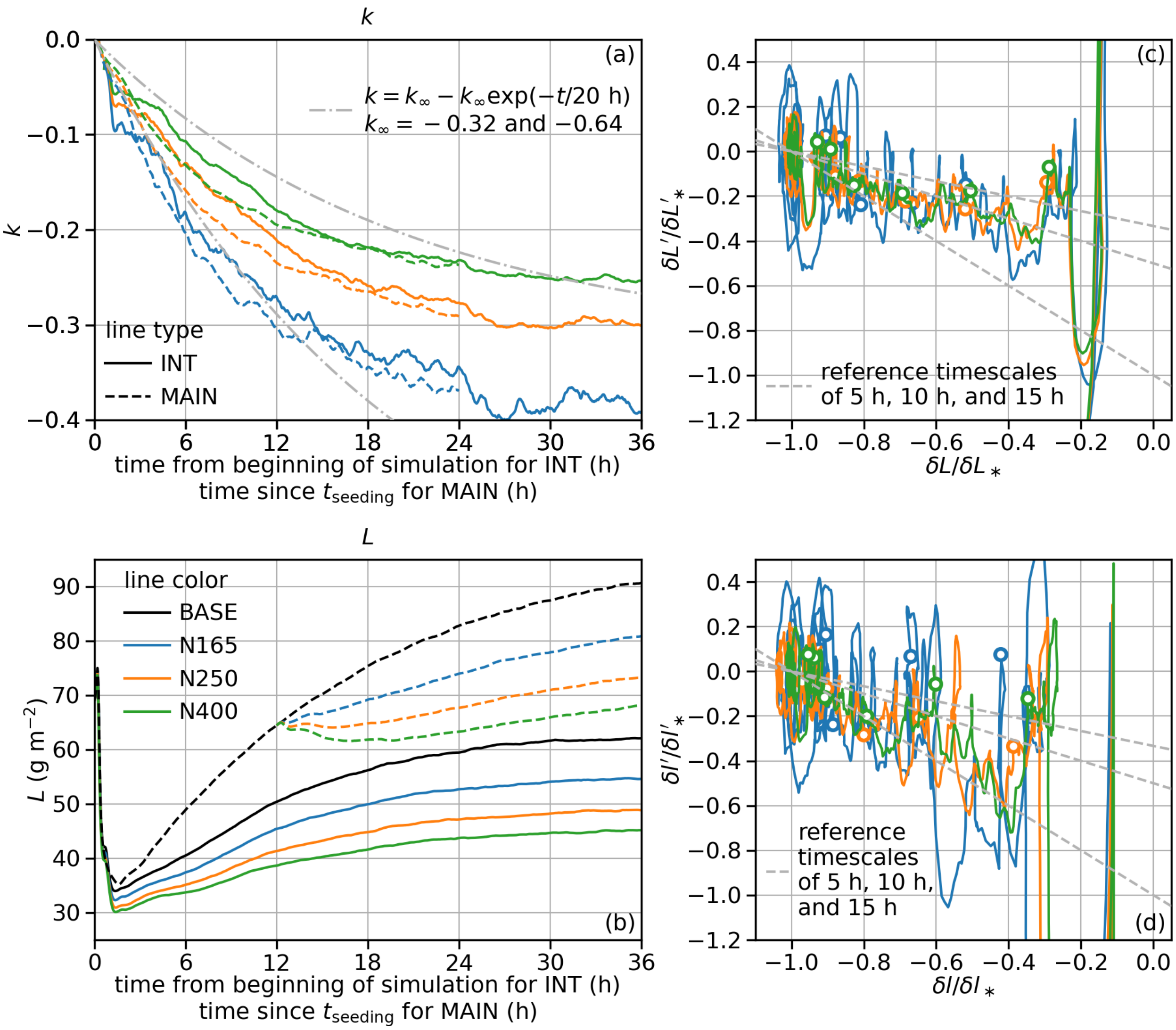

**Figure 11.** Results from the INT set averaged across 19 MCs: time series (a) $k$ and (b) $L$ in the context of results from MAIN averaged across the same 19 MCs; trajectories in (c) the $\delta L/\delta L_*$–$\delta L'/\delta L'_*$ plane and the $\delta l/\delta l_*$–$\delta l'/\delta l'_*$ plane. In Panels (c) and (d), all trajectories start from the beginning of simulations; circles are 4 h apart from 4 h to 24 h.

to MAIN. For each MC, there is a BASE run using initial $N_\text{a}$ that is not greater than BASE in MAIN ($L$ in INT is overall lower than in MAIN and hence can support lower $N$ without precipitating); then there are three perturbation runs where the initial 470 $N_\text{a}$ is setup to achieve similar $N$ to N165, N250, and N400 runs in MAIN. Each simulation lasts for 36 h. One MC in INT has maximum cloud top hitting the damping layer, while two other MCs have $N$ in BASE higher than the target for N165 at the end of 36-h simulations. These three MCs are removed and we focus on the results from the remaining 19 MCs.

Figure 11a shows the $k$ time series from this INT set. It is more negative for lower $N$, as in MAIN. In the first 2 h, $k$ between BASE and the seeded run is clearly negative and different between seeded runs, suggesting that $N$ leaves a footprint on the very first few overturnings in the BL, even though the turbulence during that time is not realistic. This is also evident in the $L$ time series in Figure 11b. However, by 24 h from the beginning of the simulation, $k$ is comparable to those in MAIN at 24 h since $t_{\mathrm{seeding}}$, suggesting that the less negative $k$ at high $N$ is not a result of compensation of increased entrainment by surface flux, which is consistent with results in Section 3 that the response in the surface flux is weak.

The evolution of $k$ is slower in INT than in MAIN, evident in that the $k$ from INT are less negative than their counterparts from MAIN before they become similar around 24 h. Still, the timescale for $k$ is shorter than 20 h, given that their time series are more curved than reference curves representing exponential convergence of $k$ with a 20-h timescale (Figure 11a).

Figure 11c shows the ($\delta L$, $\delta L'$) trajectories from INT. After the initial few hours with dramatic fluctuation, $\delta L$ evolves with a long $\tau_{\delta L}$, which arguably becomes shorter after 12 h from the beginning of the simulation. This slow development may be related to the overall weak turbulence in BLs with low $L$ (Figure 11b) and partial cloudiness. (The cloud fraction, $f_{\mathrm{c}}$, defined as the fraction of domain with cloud optical depth greater than 1, is around 74% at 2 h from the beginning of the simulation, increases to 92% at 12 h and continue to increase towards 36 h. Not shown.) However, recall that it is the timescale for $\delta l$, not $\delta L$, that directly relates to the timescale of $k$. For INT, the timescale for $\delta l$ is shorter than $\delta L$ (Figure 11d), supporting a shorter timescale for $k$.

### 4.5 Implications

Proposed MCB efforts involve the seeding of large areas of marine stratocumulus using arrays of aerosol sprayers (Wood, 2021; Feingold et al., 2024). Our results suggest that the $L$ in nocturnal, non-precipitating marine stratocumulus has the highest negative $L$ response per unit amount of $N$ perturbation on a log scale when seeding is applied to cloud fields with low background $N$, i.e., close to but above the value of $N$ associated with the onset of precipitation. Still, it is unlikely that the negative $L$ adjustment at these values of $N$ will fully offset the brightening due to the Twomey effect during the course of one night (assumed to be 12 h long). With increasing $N$, the entrainment enhancement and therefore negative $L$ adjustment becomes less efficient. As a result, stronger $N$ perturbations would likely benefit MCB not only via the Twomey effect but also by reducing the magnitude of the negative adjustment slope. Recently, Prabhakaran et al. (2024) suggested that if non-precipitating marine stratocumuli were to be chosen as targets for MCB, it would be preferable to inject as much aerosol as possible, up until the point that aerosol particle coagulation losses start to dominate. This result is in accord with the results presented in the current work.

The development of the negative $\delta L$ is a manifestation of the feedback between initially enhanced cloud-top entrainment, $L$, and BL turbulence, soon after the seeding stops. In our simulations, this process could follow a short timescale (around 5 h) for 8–12 h. Our results are robust for STBLs that are close to well-mixed and overcast without shear as a primary source of BL turbulence. It remains to be seen whether additional sources or sinks of BL turbulence, e.g., shear (Kazil et al., 2016) or the presence of a subcloud stable layer (Zhang et al., 2023), and additional timescales introduced by other processes in the real

world (e.g., drift in SST and FT humidity, spreading of ship tracks) would change the timescale and duration of the $\delta L$ and $k$ evolution for real clouds.

## 5 Summary

In this study, we have explored the magnitude and the timescale of liquid water path ($L$) adjustment to cloud droplet number concentration ($N$) for nocturnal non-precipitating marine stratocumulus using large-eddy simulations (LES). For each of 22 meteorological conditions (MCs), a 36-h BASE run is performed with relatively low $N$. Then a set of aerosol perturbation runs are performed by seeding the BASE at $t_{\mathrm{seeding}} = 12\,\mathrm{h}$ and extending the simulations until 36 h. All simulations feature boundary layers (BLs) that are close to well-mixed and overcast, without shear as a primary source of BL turbulence.

The main findings include:

- The $L$ adjustment ($\delta L$) slope ($k$) is more negative for simulation pairs with relatively low $N$ and less negative for high $N$, consistent with Lu and Seinfeld (2005) and Chen et al. (2011). This result agrees with the expectation that the effects of several mechanisms for entrainment enhancement by increasing $N$ all saturate at high $N$. Overall, a $k$ more negative than $-0.4$ is unlikely within 24 h since $t_{\mathrm{seeding}}$.

- The evolution of $\delta L$ follows a short timescale. To be specific, the $\delta L$ tendency quickly becomes negative as seeding starts. Soon after seeding ends, it decays towards 0 as $\delta L$ continues to become negative following approximately an exponential convergence that is governed by a short timescale of around 5 h. The evolution follows this exponential convergence for around 8–12 h.

- Like the short timescale investigated by Jones et al. (2014), our short timescale emerges as a result of the feedback between entrainment, $L$, and boundary layer (BL) turbulence (e.g., Zhu et al., 2005; Bretherton and Blossey, 2014) driving the $\delta L$ evolution. To be specific, the seeding enhances the entrainment velocity, which reduces $L$ and weakens the boundary layer (BL) turbulence, leading to the decay of the entrainment velocity deviation in seeded runs from BASE while the responses in entrainment efficiency and integrated buoyancy flux persist. This feedback dominates because other processes, including the radiation, surface fluxes, and subsidence, only respond to the seeding weakly.

- This short timescale is insensitive to the amount of seeding. As a result, the evolution of the deviation of several quantities in seeded runs from BASE is similar after appropriate scaling. The other parameter governing the exponential convergence of $\delta L$, namely the initial $\delta L$ tendency, is sensitive to the amount of seeding.

- The timescale of $k$ evolution is closely related to the timescale of $\delta L$ and hence also short, while it could also be affected by the timescale of $\delta N$.

In Figure 1, we also observe some correlation between $k$ and meteorological conditions given similar $N$, like many previous works did. This dependence will be examined in detail in a future study. For the rest of the paper, we have presented results

based on multi-MC composite for different aerosol configurations. This is because the time series for individual simulations or individual simulation pairs are noisy, especially when it comes to $k$ and $k'$. The compositing masks the variation among MCs in terms of both the magnitude and timescale of $\delta L$ and $k$ evolutions, which should be examined in the future.

Even though LES is advantageous for resolving some small-scale processes, quantifying the effects of individual mechanisms behind the N-enhancement of the entrainment using LES takes very careful expriment design (e.g., Igel, 2024) and would be computationally very expensive to do for a wide range of conditions. A companion paper led by Hoffmann et al. (2024) addresses this issue in a MLM framework.

*Code and data availability.* The System for Atmospheric Modeling (SAM) code is publicly available at http://rossby.msrc.sunysb.edu/SAM/. Data for reproducing the results will be provided following acceptance.

*Author contributions.* YC and PP conceived the study. YC designed and performed the simulations with suggestions from PP and FH. YC led the data analysis with critical insights from PP and JK. YC wrote the manuscript. All authors contributed to the interpretation of the results and provided comments on the manuscript.

*Competing interests.* At least one of the (co-)authors is a member of the editorial board of *Atmospheric Chemistry and Physics*. Other than this, the authors declare that they have no conflict of interests.

*Acknowledgements.* This study has been supported by the U.S. Department of Energy (DOE), Office of Science, Office of Biological and Environmental Research, Atmospheric System Research (ASR) program (Interagency Award Number 89243023SSC000114); the U.S. Department of Commerce (DOC); the National Oceanic and Atmospheric Administration (NOAA) Climate Program Office as part of the Earth's Radiation Budget (ERB) program (award no. 03-01-07-001); and NOAA cooperative agreements (grant nos. NA17OAR4320101 and NA22OAR4320151). Fabian Hoffmann appreciates support by the Emmy Noether Program of the German Research Foundation (DFG) under grant no. HO 6588/1-1. The computational and storage resources are provided by the NOAA Research and Development High Performance Computing Program (https://rdhpcs.noaa.gov, last access: 27 November 2024). We thank Marat Khairoutdinov for graciously providing the SAM model and Wayne Angevine for a few valuable discussions.

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
