# Peer review of "Magnitude and timescale of liquid water path adjustments to cloud droplet number concentration perturbations for nocturnal non-precipitating marine stratocumulus"

_EGUsphere, 2024_

## Author Response (AR1)

We thank the reviewers for their constructive comments.

In the rest of this file, we show the comments from Reviewers #1 and #2 in red and blue, respectively, and our responses in black.

After addressing reviewers' comments, we list some minor edits we have made during the revision.

_Some changes are not tracked to avoid cluttering the tracked-changes file. They are noted in italics with underline._

**Reviewer #1**

https://doi.org/10.5194/egusphere-2024-3891-RC1

I find this to be an outstanding manuscript regarding the timescales of LWP adjustments in non-precipitating marine stratocumulus clouds. I really have no substantial suggestions for improvement.

1. Perhaps all prime quantities could be written as dot quantities (symbol with a dot overhead) instead? This is the more typical shorthand notation for time tendencies in the physics community I believe.

The notation has been changed in both the text and the plots. _Since there are more than 100 occurrences, we decide not to track these changes to avoid cluttering the tracked-changes file._

The text defining the notation has been modified accordingly. Please see L51 in the revised manuscript or L51 in the tracked-changes file.

2. A reference for the -0.4 value would be appreciated.

We have added a reference to the specific equation to support our interpretation of $k$ being $-0.4$. Please see L32–33 in the revised manuscript or L33 in the tracked-changes file.

3. Line 260 - typo: So --> To

This typo has been fixed. Please see L264 in the revised manuscript or L264 in the tracked-changes file.

**Reviewer #2**

https://doi.org/10.5194/egusphere-2024-3891-RC2

Review of "Magnitude and timescale of liquid water path adjustments to cloud droplet number concentration perturbations for nocturnal non-precipitating marine stratocumulus" by Chen, Prabhakaran, Hoffman, Yamaguchi, Kazil and Feingold, Manuscript egusphere-2024-3891

Recommendation: Minor revisions

This nice paper revisits the earlier work of many of the coauthors in Glassmeier et al (2021, Science), which suggested a strong thinning of non-precipitating stratocumulus in response to increased aerosol concentrations for nocturnal-only simulations in the limit of long time behavior. Here, a subset of cases from Glassmeier et al has been selected, redesigned to include a set of aerosol perturbations (seeding) for each case and other changes, re-simulated and re-analyzed. The main result is that a shorter entrainment-related adjustment timescale seems to dominate the liquid water path evolution of the clouds up to about 12 hours after seeding. The analysis is clear and well-presented, as suggested by the relatively few suggestions for wording changes (which is unusual for me). My comments ask somewhat technical questions about how to think about approaching equilibria when multiple adjustment timescales exist. I also question whether it might be possible to conclude the paper in a stronger way, as I found section 4 (while interesting) distracted a bit from the main story of the paper.

=========================

Major comments:

1. (14/289) I am somewhat uncomfortable with equation 19 as stated, because (1) (as the paper very clearly states) there are multiple timescales involved in the adjustment to equilibria and (2) that it is difficult to identify L_\infinity without actually running to equilibrium over very long timescales (days to 10s of days). Could some language be added to make clear that the diagnosed value of \tau is only valid locally in time and may well evolve with the state of the system? Also, is the L_\infinity here really the equilibrium value of LWP, or is it more representative of a point on the slow manifold (maybe L_qe for quasi-equilibrium) where the fast timescale has adjusted but the slower adjustment timescales are still at work?

We meant to confirm here that the evolution of $L$ in simulations of 36 h duration can be approximately fit with an equation with one timescale. We have updated the text to make this clear. The use of "$L_\infty$" is to be consistent with Glassmeier et al. 2021 but we now first make it clear that $L_\infty$ (as well as $\tau$) is a parameter coming out of the fit and then interpret them as an apparent steady state and the inverse of a timescale.

Please see Section 3.4 in the revised manuscript or L291–302 and L315–316 in the tracked-changes file.

In Section 4.1 (Comparison of $\delta L$ timescales with previous studies), we now introduce a term called "equilibrium state" to refer to model states that "are in equilibrium with the boundary conditions and forcings in the simulations". Following Bretherton et al. 2010JAMES, the "equilibrium" here

emphasizes the balance between the entrainment velocity and the subsidence velocity.

$$w_e + w_s = 0. \tag{1}$$

Then it becomes clear that our model states are approaching some apparent steady states and have not reached their equilibrium states.

The timescale is also apparent, as the reviewer pointed out. However, this is the case not only in our simulations but also the previous works by Bretherton et al. 2010JAMES and Jones et al. 2014JAMES, unless the governing equation for the system allows a clean separation of the timescales associated with different processes (e.g., the linearized MLM without certain forcing terms in Jones et al. 2014JAMES) or when there is only one process left (e.g., the late stage of evolution in Bretherton et al. 2010JAMES). In other words, there is no "clean" timescale in our paper to distinguish from. So, we omit "apparent" here. However, we now explicitly mention that the timescale is the result of various processes, each with its own timescale, in Section 4.1.

Please see Section 4.1 in the revised manuscript or in the tracked-changes file. We have also (1) changed "steady state" to "equilibrium state" and (2) changed "steady state" to "apparent steady state" in a few other places, changes tracked but specific line numbers omitted here.

2. The material in section 4 is very interesting for me, but I found that it distracted from the flow of the paper to its conclusion. Might some of the sensitivity studies be moved to one or more appendices, so that the main body of the paper might be more focused? The authors may have a different view, but I worried that the size/variety of material in the discussion might affect the impact of the paper on its readers.

Thanks for pointing this out. Regarding the original Sections 4.3 (A few sensitivity tests), we have

- Moved the first test to Appendix A. (This test showed that $k$ between $N$165 and 250, both seeded from BASE was comparable to $k$ between the new BASE runs where $N$ were close to those in $N$165 and new seeded runs where $N$ were close to those in $N$250.)
- Moved the sensitivity to $N$ evolution to Appendix B.
- Moved other sensitivity tests to Appendix C.

In addition, we have moved Section 4.4 (Impacts of perturbing initial $N_a$ and fixed surface flux) to Appendix D. _The moving of these two sections is not tracked. But additional edits are._

We now

- Refer to Appendix A in Section 3.1 (Overview). Please see L193–195 in the revised manuscript or L193–195 in the tracked-changes file.
- Refer to Appendix B in Section 3.3 ($k$ budget). Please see L281–282 in the revised manuscript or L281–282 in the tracked-changes file.
- Refer to Appendices C and D when we describe the applicability of the main results in Section 4.3 (Implications, the original Section 4.5). Please see L460–461 in the revised manuscript or L469–470 in the tracked-changes file.

==========================

Specific/minor comments (11/240 means p. 11, line 240):

14/294: I would speculate that some of the oscillations in \delta L and \delta L' after seeding result from horizontal heterogeneities of aerosol and cloud droplet number in the cloud layer. With seeding happening below 400m, the increased aerosol would be carried into the cloud layer in updrafts and spread horizontally over time. If frequent outputs were not saved after seeding, this may be unknowable. If horizontal heterogeneity is driving some of the oscillations, you might consider citing Dhandapani et al (2025, https://doi.org/10.1029/2024MS004546) which talks about the preferential lofting of seeded aerosol in updrafts. If the authors know a better/earlier reference, that might also be helpful. The origin of this comment is partly that the timescale analysis is partly based on the assumption that the LES is behaving more or less like a mixed-layer model, and the horizontal heterogeneities would deviate from the well-mixed assumption in the same way that a decoupled boundary layer would in the vertical.

This is a good point. We now include this horizontal heterogeneity in aerosol as a potential mechanism for the oscillation and make a reference to Dhandapani et al. 2025. Please see L311–313 in the revised manuscript or L312–314 in the tracked-changes file.

15/305: Regarding: "Considering that L for BASE is approaching its steady state, this means that L for N400 is probably approaching a different steady state." Is it really obvious that the different simulations are approaching different steady states? They're certainly on different slow manifolds because w_e is a function of both buoyancy flux and Nd. However, the N perturbation is steadily decaying over time. My guess is that all of the seeded simulations would converge to the long-term-equilibrium solution of BASE. For this not to be true would require a solution where the higher-Nd entrainment would match the subsidence rate at a particular inversion height where the aerosol budget is also at equilibrium (with the surface source balancing changes due to entrainment and collision-coalescence). However, this would only apply if the boundary conditions (i.e., free tropospheric temperature and moisture profiles) remain fixed, which seems to not be true in the default setup here.

While I've raised this question, I wouldn't encourage the authors to spend a lot of computer time trying to figure this out (though it could be done pretty easily in a mixed-layer model that includes prognostic aerosols, using the entrainment parameterization in the companion Hoffmann et al paper). My suggestion would be to think about whether this would impact ideas about the value of k_\infiny. My view is that the quasi-equilibrium value of k (maybe k_qe) computed between points on different slow manifolds (with different Nd) are probably well defined and more meaningful than the true equilibrium value since the equilibrium will take a very long time to reach.

We share a lot of the thoughts and speculations with the reviewer. The $N$ in the high $N$ runs were still decreasing at the end of the simulations, as pointed out near the end of Section 3.1 in the original manuscript. We agree that the long term evolution of $N$ will have an impact on the equilibrium states. And since we only seeded over a limited period of time, boundary conditions (surface fluxes and mixing with FT air) will shape the longterm $N$.

In this paper, we did not try to address the equilibrium states. Instead, we described the results based on our 36-h simulations, and then showed where they fit into the conceptual picture of the long-term evolution that approaches the equilibrium states.

The statement the reviewer refers to came up after we presented the results that $\delta \dot{L}$ approached 0 and $\delta L$ stabilized. We agree that here we should also make it clear that $\delta L$ may become weaker

due to the long-term evolution of $N$. We have updated the text to remove this statement. We have also added some discussions in Section 4.1 when we discuss the equilibrium states. Please see the end of Section 4.1 in the revised manuscript or L403–404 in the tracked-changes file.

17/345: Please write these equations down explicitly, either here or in the supplement. It's too much to expect the reader to reconstruct them.

The ODE set has been explicitly presented. Please see L354–359 in the revised manuscript or L356–362 in the tracked-changes file.

17/355-360: The "intentional"/"unintentional" language seems odd to me, though maybe that reflects my background. In my mind, "unintentional" = "perturbed boundary conditions" and "intentional" = "perturbed model state" or "perturbed initial conditions". Hopefully, the reader should understand that perturbed BCs are likely to result in a different equilibrium solution while perturbed ICs will not, but it may be useful to remind the reader of this.

We used "perturbation" to refer to the deviation from the equilibrium state. So, when we introduced Bretherton et al. 2010JAMES, "perturbation" referred to the mismatch between the initial state and the equilibrium state. It did not mean the initial state perturbation, e.g., using different initial $z_i$. In contrast, Jones et al. 2014JAMES let their MLM and LES reach equilibrium states first, and then introduced perturbation. So, their perturbation is the same as the deviation from the equilibrium state.

With this clarification, hopefully it makes sense now that the perturbation in Jones et al. 2014JAMES is "intentional" because the experiment design was to introduce perturbations to reveal all timescales. We agree that using "unintentional" when introducing Bretherton et al. 2010JAMES could be confusing. There, the "intentional" perturbation from the equilibrium state is still needed to examine timescales as the system approaches the equilibrium state. However, since the focus in that paper was the thermodynamic timescale ($\tau_{th}$) and the divergence/subsidence timescale ($\tau_i$), the short timescale was not part of the focus and emerged "unintentionally".

Looking back, we agree that whether the perturbation or the emergence of the fast timescale in those two papers was "intentional" is not something that we would like to distract the readers with. We have modified the text to clean this up.

We agree with the reviewer that perturbing forcings (boundary conditions and others) may change the equilibrium state while perturbing states (initial or not) does not. However, we would like to clarify that this is not the focus of the discussions in this subsection.

We have modified the text accordingly. Please see Section 4.1 in the revised manuscript or in the tracked-changes file.

20/433: Regarding "... their L may carry the "memory" of different N." I don't quite understand what's being discussed here. Is it just that the state of the boundary layer (zi, Nd, etc.) is different than the BASE case because it went through a period of intensified entrainment after seeding? Or does a quantity like entrainment depend on both the present and past values of Nd? The latter seems unlikely to me. If it's the former, I don't think the "memory" effect is meaningful, because L and N aren't a complete specification of the state of the system, so of course two simulations with identical L and N could vary because (even for a mixed-layer model) there are two more

state variables that constrain the system. This reliance on a limited number of state variables (or quantities derived from state variables like L) also hides some of the disequilibrium of the system and is related to my comment about L_\infinity above.

For two simulations, first one without seeding and the second one with seeding but $N$ nudged towards the final $N$ in the previous simulation,

$$L_1(t) = L_0 + \int_0^t \dot{L}_1(t) \tag{2}$$

$$L_2(t) = L_0 + \int_0^t \dot{L}_2(t) \tag{3}$$

What we meant was, $L_2(t)$ will not be the same as $L_1(t)$ even if the $N$ in both simulations are very close at the end of the simulations. This has a small impact on our results regarding $k$ dependence on $N$ (i.e., $k$ between low $N$ pairs could be a bit more negative), but it does not affect the overall conclusion.

We have updated the text to avoid the use of "memory". We have also made some changes in wording to emphasize that we are talking about BASE and seeded runs in MAIN and this sensitivity test, not just two simulations with the same $(N,L)$ at some point. Please see Appendix B in the revised manuscript or in the tracked-changes file.

23/498-500: Are there any worries that seeding-induced decoupling might lead to cloud thinning, independent of the LWP decreases in the mostly well-mixed boundary layers seen in this work?

We now emphasize that both BASE and seeded runs in our simulations are well-mixed. Please see L534 in the revised manuscript or L549 in the tracked-changes file. Also see L459 in the revised manuscript or L468 in the tracked-changes file.

24/506: Please add "diurnal cycle" to the list in parenthesis. It's arguably the most important outside timescale that's neglected in the simulations in this paper.

We agree with this assessment and "diurnal cycle" has been added as one source of additional timescale. Please see L463 in the revised manuscript or L472 in the tracked-changes file.

Other than the changes required to address the reviewers' comments, we have also made the following edits.

- We have removed "$(\mathrm{d}I/\mathrm{d}n)$" from L29 in the original manuscript because this expression is never used in the rest of the paper. Please see L29 in the tracked-changes file.
- In L383 in the original manuscript, we incorrectly used $\delta w_\mathrm{e}'$ to represent the scaling parameter for $\delta w_\mathrm{e}$. It has been changed to $\delta w_\mathrm{e}^*$. Please see L407 in the revised manuscript or L413 in the tracked-changes file.
- In Sections 4.2 (Similarity of $\delta L$ evolution) and 4.4 (Impacts of perturbing initial $N_\mathrm{a}$ and fixed surface flux, currently Appendix D) in the original manuscript, some scaling parameters were denoted with subscript "$*$" while others with a superscript "$*$". For example, $\delta L'$ was scaled with $\delta L_*'$ while $\delta \zeta_\mathrm{c}$ with $\delta \zeta_\mathrm{c}^*$. We now consistently use a "$*$" as a superscript to denote the scaling parameters. _These changes are not tracked to avoid cluttering the tracked-changes file._
- The panel title of Figure 5b has been simplified from "$\dot{I}(=\dot{L}/L)$, 4-h smoothed" to "$\dot{I}=\dot{L}/L$, 4-h smoothed".
- In addition, we have made some minor wording changes. Please see L213, L314, and L442 in the tracked-changes file.
- We have updated author affiliations and acknowledgment according to the latest guidance from the authors' institute.